

# Stable and unstable fall motions of plate-like ice crystal analogues

Jennifer R. Stout[1], Christopher. D. Westbrook[1], Thorwald. H. M. Stein[1], and Mark W. McCorquodale[2]

[1]Department of Meteorology, University of Reading, Reading, UK
[2]Department of Civil Engineering, University of Nottingham, Nottingham, UK

**Correspondence:** J. Stout (j.r.stout@pgr.reading.ac.uk)

**Abstract.**

The orientation of ice crystals affects their microphysical behaviour, growth, and precipitation. Orientation also affects interaction with electromagnetic radiation, and through this, influences remote sensing signals, in-situ observations, and optical effects. Fall behaviours of a variety of 3D-printed plate-like ice crystal analogues in a tank of water-glycerine mixture are observed with multi-view cameras and digitally reconstructed to simulate falling of ice crystals in the atmosphere.

Four main falling regimes were observed: stable, zigzag, transitional, and spiralling. Stable motion is characterised by no resolvable fluctuations in velocity or orientation, with the maximum dimension oriented horizontally. The zigzagging regime is characterised by a back-and-forth swing, corresponding to a time series of inclination angle approximated by a rectified sine wave. In the spiralling regime, analogues consistently incline at an angle between 7 and 28 degrees, depending on particle shape. Transitional behaviour exhibits motion in between spiral and zigzag, similar to that of a falling spherical pendulum.

The inclination angles that unstable planar ice crystals make with the horizontal plane are found to have a non-zero mode. This observed behaviour does not fit the Gaussian model of inclination angle that is common in the literature. The typical Reynolds number when oscillations start is strongly dependent on shape: solid hexagonal plates begin to oscillate at Re = 237, whereas several dendritic shapes remain stable throughout all experiments, even at Re > 1000.

## 1 Introduction

Understanding the motion of falling ice crystals is important to both the microphysical processes within clouds and their bulk characteristics, such as radiative and optical properties. However, their dynamics are not well understood; ice crystals have complex and irregular shapes, and can produce fluttering, spiralling, and tumbling trajectories.

To quantify the orientation of analogues, the inclination angle, $\theta$, is the angle made between the rotated ice crystal's c-axis and the global vertical y-axis (Figure 1). Falling ice crystals, when stable, have a constant $\theta$ of 0° (List and Schemenauer, 1971). When unstable, it is commonly assumed crystals have Gaussian distributions of orientations, with a modal $\theta$ of 0°, and standard deviations varying between 10° (pristine ice crystals) and 40° (heavily aggregated snowflakes) (Melnikov and Straka, 2013; Ryzhkov et al., 2020).





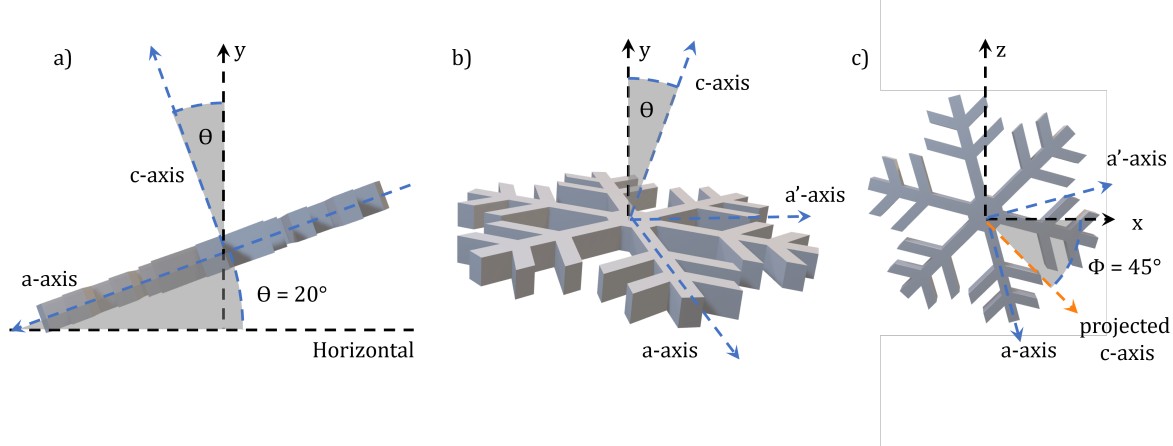

**Figure 1.** Axes of a crystal inclined by theta of 20° and pointing towards an azimuth, $\phi$, of 45°, both relative to the lab frame of reference, given by an (x,z) horizontal plane and a vertical y-axis. The crystal plane is represented by the a and a'-axis, where the a-axis is aligned with one of the crystal arms. The c-axis is perpendicular to the crystal plane. The views provided are for an observer facing (a) parallel to the c-y plane (b) parallel to the a-y plane (c) parallel to the x-z plane.

## 1.1 Importance of Orientation of Ice Crystals

The orientations of falling crystals impact their projected area in the horizontal plane, their sedimentation rate, and the rate at which they can collide with other hydrometeors (Westbrook et al., 2010). Compared to ice crystals with purely vertical motion, ice crystals with horizontal motions in addition to the vertical will travel a farther distance, providing more opportunity to collide with other hydrometeors than ice crystals with vertical motion alone (Wang, 2021). This further impacts cloud macrophysical properties, such as radiative impacts and cloud lifetime.

Properties of ice crystal motion have important implications for radar and lidar observations: orientation directly influences signals sampled by dual-polarisation radar, as the orientation of crystals changes the Differential Reflectivity ($Z_{DR}$) (Bringi and Chandrasekar, 2001). Unstable motion causes fluctuations in the crystal velocity in the component of the crystal motion along the radar beam, broadening the Doppler spectrum width (Feist et al., 2019). Differences in assumptions of orientation can therefore impact the relationship between derived ice crystal diameter and doppler or polarimetric remote sensing observations,
ultimately affecting radar-derived precipitation rates (Matrosov, 2011; Schrom et al., 2023).

Horizontal orientation of ice crystals affects lidar observations, especially in the case of specular reflection, causing enhanced return for lidars pointing exactly at zenith or nadir (Sassen, 1977; Platt, 1977; Gibson et al., 1977; Hogan and Illingworth, 2003). The magnitude of the enhancement and its variation with elevation angle are strongly dependent on the chosen model for crystal orientation Platt (1977).

For clouds containing ice, crystal size, concentration, habit, and orientation all play a significant role in determining cloud radiative properties such as optical depth and albedo (Curry and Ebert, 1992; Ishimoto et al., 2012; Hirakata et al., 2014).



Changes in these particle orientation assumptions can lead to high variation in the retrieval of cirrus properties from satellite observations. In certain cases, decreasing the assumed standard deviation of $\theta$ from 20° to 5° doubled the estimated optical depth (Masuda and Ishimoto, 2004). Horizontally oriented ice crystals have also been theorised to increase cloud shortwave albedo by up to 40% (Takano and Liou, 1989).

When ice crystals are horizontally oriented, this gives them distinctive optical characteristics (Cho et al., 1981; Sassen, 1987). For instance, horizontal crystals can create a range of atmospheric optical phenomena such as sun dogs, light pillars, and Parry arcs, among others (Moilanen and Gritsevich, 2022). Additionally, spiralling ice crystals have been hypothesized to cause the rare 'Bottlinger's rings' effect (Lynch et al., 1994; Tränkle and Riikonen, 1996).

Ice crystal orientation also impacts the apparent crystal properties (e.g. size, projected area, aspect ratio) inferred from analysis of 2D projections sampled by ground-based imagers such as PIP (Jiang et al., 2017; von Lerber et al., 2017). To estimate the three-dimensional parameters relevant for drag calculations from two-dimensional projections of snowflakes assumptions about particle orientation, shape, motion, must be made (Köbschall et al., 2023). Dunnavan and Jiang (2019) find that for highly eccentric particles (such as aggregates) that have large fluctuations in $\theta$, very limited information can be inferred about a particle's 3D shape without specifying appropriate particle orientation distributions.

## 1.2 Phenomenology of Circular Discs

Analogies may be drawn between the aerodynamics of ice crystals and those of other idealized shapes, such as thin discs. There has been extensive experimental research on the aerodynamic behaviour of planar discs (e.g. Willmarth et al., 1964; Field et al., 1997; Ern et al., 2011; Zhong et al., 2011). Two dimensionless ratios have been proposed to characterise the motion of falling circular discs: the Reynolds number, Re, and the dimensionless moment of inertia, I* (Willmarth et al., 1964; Field et al., 1997), discussed in the following subsections.

### 1.2.1 Reynolds Number

The Reynolds number is defined as:

$$Re = \frac{uD}{\upsilon} \tag{1}$$

where $\upsilon$ is the dynamic viscosity of the fluid, $u$ is the mean vertical velocity of the particle, and $D$ is the maximum dimension of the particle.

Willmarth et al. (1964) identified that Re 100 - 200 is the critical point for the onset of unstable motions for circular discs, after which periodic behaviour begins. The value of the critical Reynolds number varies depending on particle shape. Field et al. (1997) report an experimental study of how metal circular discs fall through water and glycerol mixtures, and how paper discs fall through air. Different falling regimes were observed depending upon the experimental parameters; discs could fall steadily, exhibit oscillating periodic motions, or tumble.

Periodic behaviour includes zigzag and spiralling sub-types of behaviour, and more recently, an in-between behaviour was identified as transitional, through experimental investigations by Zhong et al. (2011). Zhong et al. (2011) also finds that motion





typically becomes more planar (more likely to zigzag) for increasing Reynolds number and dimensionless moment of inertia,

but the trajectory becomes more circular (more likely to spiral) for decreasing Reynolds number and dimensionless moment of inertia (Figure 2).

At Reynolds numbers below the critical Reynolds number, flow around crystals is stable. For planar crystals in this regime, laboratory and field measurements have shown that the largest dimension becomes normal to the axis of gravity, and the plate crystals achieve a horizontal orientation, corresponding to a constant inclination angle of zero (Jayaweera, 1965; List and

Schemenauer, 1971; Pruppacher and Klett, 1997; Kajikawa, 1992).

At a critical Reynolds number, the flow around the crystal becomes unstable, forming vortices as part of the boundary layer of fluid at the surface of the particle. When shedding of these vortices in the wake of crystal occurs, the distribution of pressure on the crystal changes, exerting forces that cause it to rotate (Zhong et al., 2013; Tagliavini et al., 2021). These unstable motions are observed as oscillations in orientation, as well as the vertical and horizontal velocities, such that they are non-zero,

fluctuating, and have a distribution. There is a current lack of understanding about the orientation of ice crystals in unstable regimes, and one of the aims of this paper is to explore this.

### 1.2.2   I*

The dimensionless moment of inertia for a thin disc is defined as the ratio of the moment of inertia of a thin disc about its diameter and a quantity proportional to the moment of inertia of a rigid sphere of fluid of the same diameter, such that:

$$I^*_{disc} = \frac{\pi}{64} \frac{\rho_P}{\rho_f} \frac{t}{D} \tag{2}$$

where $\rho_P$ is the density of the particle, $\rho_f$ is the density of the fluid, $t$ is the thickness of the disc, and $D$ is its diameter (Willmarth et al., 1964). For more complex shapes, the more general non-dimensional moment of inertia is:

$$I^* = \frac{I_a}{\rho_f D^5} \tag{3}$$

where $I$ denotes the principal moments of inertia of a particle with maximum dimension, $D$. The moment of inertia for rotation

around the 3 principal axes of the crystals is calculated, where $I_a$ is the smallest of these three moments, aligned in the a-axis of the crystal (Fig. 1) (Kajikawa, 1992).

### 1.2.3   Comparing ice crystals to discs: Re - I* Phase Space and Area Ratio

Figure 2 presents a summary of the coverage of the data presented in McCorquodale and Westbrook (2021a, b), also used in this study, throughout Re and I* phase space, as well as the key prior experiments on ice crystal shapes (Kajikawa, 1992;

Cheng et al., 2015; Nettesheim and Wang, 2018) and circular discs (Field et al., 1997; Zhong et al., 2011) that are discussed in sections 1.3 and 1.4 . Using a mass-diameter relationship from Nakaya and Terada (1935) for planar dendritic crystals, and methods for estimating I* from Kajikawa (1992), we find that a 10 mm, 1 mm, and 0.1 mm planar dendritic crystal where the density of ice is 917 $kgm^{-3}$, and the density of air is 1.2 $kgm^{-3}$, have an I* of 0.02, 0.2, and 2.0 respectively. Figure 2 displays



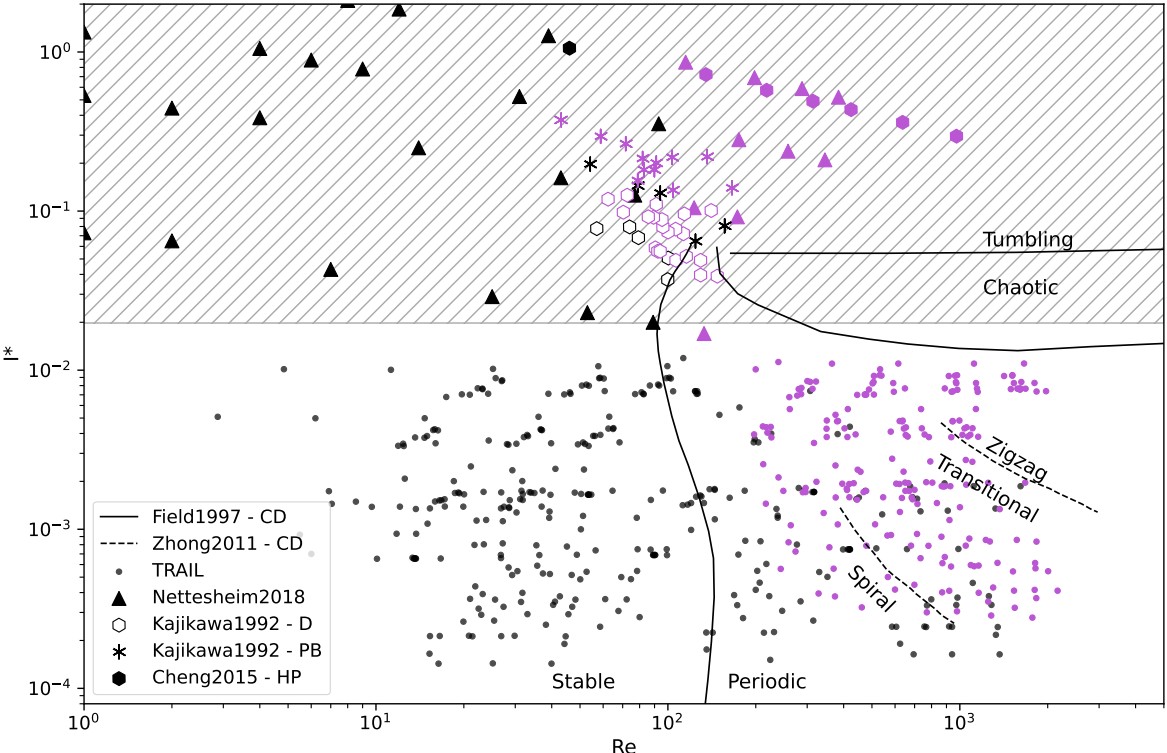

**Figure 2.** Phase diagram showing the stable (black) and unstable (purple) behaviour of falling particles as a function of I* (dimensionless moment of inertia) and Re (Reynolds number). Data from TRAIL (this study and McCorquodale and Westbrook (2021a, b)) is in solid circles, and all other data points are for shapes relevant to ice crystals, (Nettesheim and Wang, 2018; Kajikawa, 1992; Cheng et al., 2015). Solid lines and annotations are from Field et al. (1997), dashed lines and rotated annotations are from Zhong et al. (2011), presenting the observed behaviour for circular discs. Hatched region is the expected range of I* for dendritic planar ice crystals.

this expected range of I* for planar dendritic ice crystals — this matches up well with the range of previous observations of ice

crystals.

   Our study focuses on planar crystals. These range from hexagonal plates (which present a solid obstacle to the flow at all Reynolds number), to stellar crystals and dendrites which have much more open projections. One way to characterise this shape variability is by area ratio; the ratio of the maximum cross-sectional area of the particle and the area of its circumscribing circle, which helps compare to circular discs. Fluid experiments on planar shapes report that the amplitude of oscillations in the

descent velocity was maximum for circular discs and decreased with area ratio, suggesting that unstable motions are inhibited by more complex shapes (Esteban et al., 2018).



## 1.3 Existing Work on Orientation of Ice Crystals

A variety of approaches have been developed to study the aerodynamics of ice crystals. Using the Cloud–Aerosol Lidar and Infrared Pathfinder Satellite Observations (CALIPSO), Zhou et al. (2013) simulated crystal distributions and orientations and found that horizontally oriented plates occurred in 60% of optically thick ice and mixed-phase cloud layers. Similarly, Stillwell et al. (2019) found that horizontally oriented plates must occur in at least 25.6% of all ice-only column observations using polarization lidar for their simulations to match the observations.

Common models of particle orientation assume either uniform distribution, horizontal orientation with inclination angle of zero, or a Gaussian distribution with a peak at zero inclination angle (e.g., Borovoi and Kustova, 2009). However, orientation models for falling particles suggest modal inclination angles of approximately 10° (Klett, 1995). Melnikov and Straka (2013) attempt to retrieve the spread of fluttering angles from polarimetric radar data, assuming that the mean inclination angle is zero, and that the distribution has a fixed, size-independent width, retrieving fluttering amplitudes on the order 2-23°. However, these indirect measurements lack the spatial resolution necessary to represent microphysical processes without making many assumptions.

More direct measurements are possible, such as in-situ observations of ice particles near the surface, (e.g., Zikmunda, 1972; Locatelli and Hobbs, 1974; Kajikawa, 1992; Garrett et al., 2015; Fitch et al., 2021). Falling natural planar snow crystals placed into a tube were studied by Kajikawa (1992), and utilising a stereophotogrammetric method found that hexagonal plate crystals exhibited stable and unstable motions, including a swing motion (zigzag), and a helical rotation motion (spiralling). The critical Reynolds number, above which crystals exhibited unstable motion, was found to vary depending on the specific crystal habit, as classified by Magono and Lee (1966). For crystals of classification P1a (hexagonal plates), the critical Reynolds number was found to be 47, while for P1f crystals (fern-like crystals), the critical Reynolds number was found to be 91. Nonetheless, the tendency of ice crystals to break, evaporate, and melt when handled led to high uncertainties in direct observations of ice crystals at the ground.

Multi-Angle Snowflake Camera (MASC) observations by Garrett et al. (2015) found that the modes of the distribution of inclination angles were 20°, 16°, and 13°, for graupel, rimed particles, and aggregates respectively, indicating that snow particles show a preference for near-horizontal orientation but have non-zero modal values. Recent research into the MASC measurements by Fitch et al. (2021) has also reported preferential non-horizontal inclinations for the orientation of snow particles, with a modal value of 12° observed for light wind speeds in shielded conditions.

These findings suggest that the assumption of Gaussian orientation distribution may not always be accurate, and that the orientation of snow particles may exhibit preferential orientations that are non-horizontal, even in quiescent environments. Lynch et al. (1994) proposed modelling falling ice crystals' swinging motion as similar to that of a pendulum where the pivot of the pendulum falls vertically at constant velocity. This notion is supported by Esteban (2019) who found that oscillatory motions of discs and a variety of other planar shapes in both quiescent and turbulent fluids had pendulum-like motions, with turbulence simply adding noise to the oscillations.



We hypothesize that planar ice crystal analogues will behave similarly to this previous experimental work, and test that hypothesis in this study. Falling ice crystals may be well approximated as falling pendulums, and there is a relationship between the distribution of angles and other fall motion aspects such as velocity fluctuations, perceived projected areas, and perceived aspect ratios.

Cheng et al. (2015) explored the behaviour of hexagonal plates using numerical simulation, with Re ranging from 46 to 974
and I* ranging from 1.1 to 0.3. The plates are stable at Re=46 and unstable at Re=135. Smaller plates exhibit a zigzag motion while larger plates exhibited spiralling, and none of the plates tumbled during the simulation, in contrast to the work by Field et al. (1997) and Zhong et al. (2011) on circular discs.

Nettesheim and Wang (2018) used numerical simulations to study the fall behaviour of branched crystals, showing unstable fall motions for sector plate at Re=384 and broad branched plate at Re=345. They also provided data on other experiments,
with sine wave fits to the Euler angles of the particles during the experiments. However, the time series the angles are fit over include a spin up period between the initial "release" of the crystal and it settling into its preferred fall motion, which precludes a quantitative comparison to the results presented in our study.

As the unsteadiness of falling particles is a complex, nonlinear, multi-degree-of-freedom phenomenon, numerical simulations impose significant computational cost and technical challenges. These simulations also rely on assumptions about
turbulence, vortex shedding, and how these interact with falling particles, making it difficult to confidently simulate the wide range of conditions ice crystals experience.

Using analogues —scaled up models of natural crystals— presents a promising avenue for studying the fall behaviour of ice crystals in a laboratory environment. List and Schemenauer (1971) report measurements of machined analogues of snowflake particles falling in solutions of water and glycerine or salt water, and exploit dynamic similarity. This dynamic similarity only
applies when falling steadily at terminal velocity, since the only dimensionless variables are $Re$ and particle shape. When falling unsteadily the ratio $\rho_p/\rho_f$, contained within $I*$, is also significant. To compare the results to natural snowflakes falling in the atmosphere. The study considers 5 different designs of planar ice crystals and observes stable behaviour at Re < 100. For discs, hexagonal plates, and broad branched models, small oscillations are observed at Re $\approx$ 200, although no oscillations are observed at this Reynolds number for stellar, dendritic, or stellar-with-plate shapes. Köbschall et al. (2023) used analogues
of aggregate snowflakes, finding that the area of complex snowflake analogues projected in the direction of flow is often maximized, and for many of their analogues, a rotation around the vertical axis was seen.

Building on previous work by Westbrook and Sephton (2017), McCorquodale and Westbrook (2021a) utilized modern 3D-printing techniques to fabricate analogues for studying the aerodynamics of ice particles through the analogue method. Experimental studies on these analogues were analysed through a custom algorithm, producing digital reconstructions of
the trajectory and orientation of the particle. From these experiments, analogues of aggregates are found to exhibit different preferential orientations depending on Reynolds number for the same particle shape (McCorquodale and Westbrook, 2021c) Tagliavini et al. (2022) performed numerical simulations with dendritic crystals, and compared results to free falling analogues, using the particle tracking algorithms described in McCorquodale and Westbrook (2021a, b). They found that throughout the Re range in both numerical simulations and laboratory observations, the wake and motions of dendritic crystals was stable,





even as high as Re = 1500, supporting the idea that the onset of unstable motions is sensitive to crystal geometry. This is a topic

explored in the current article.

### 1.4    Investigating unresolved questions

It is evident that the representation of crystal orientation in many studies is not well constrained at present. There is evidence

that unstable motions may be more complex than a simple zigzag motion, but the conditions under which this happens are not

clear.

There is extremely limited data quantifying how the orientations of unstable crystals are distributed, and what that distribu-

tion depends on, as well as how frequent and large the velocity fluctuations (in both vertical and horizontal) are in response to

the unstable wake of the falling crystal and how they are correlated with the variations in orientation. In this article we present

new data to address these areas of uncertainty.

Building on previous work by Westbrook and Sephton (2017), McCorquodale and Westbrook (2021a) utilized modern

3D-printing techniques to fabricate analogues for studying the aerodynamics of ice particles through the analogue method.

To link the behaviour of real ice crystals to the theoretical behaviour observed by Esteban et al. (2019, 2018) in laboratory

experiments, we further examine the experiments by McCorquodale and Westbrook (2021a), focusing on the fall behaviour of

quiescent plate-like particles, identify the angles at which ice crystal analogues fall and test the potential relationship between

the distribution of fall angles and other motion aspects.

The paper is organised as follows: In Section 2, we describe the experiment by McCorquodale and Westbrook and the data

sets derived from it. In Section 3 we discuss the results, beginning with section 3.1, discussing which particles fall steadily.

Section 3.2 introduces and and describes four case studies of periodic motion and how their orientations, velocities, and

oscillation frequencies can be characterised. Section 3.3 discusses the broader trends and characteristics of the full data set,

including how distributions of $\theta$, oscillation frequencies. and motion type vary by shape and Reynolds number, as well as how

velocity components vary with one another. Further discussion of these results, including summary and conclusions, can be

found in Section 4.

### 2    Data

A diverse range of ice particle analogues were included in this study, ranging from hexagonal plates with an area ratio of 0.87

to open branched crystals with area ratios as low as 0.23 (Table 1). The area ratio of the particles included is calculated using

the observed projected area of the particle divided by the circumscribing circle at each time-step during experiments when fall

motion is stable. The mean calculated area ratio is then used to describe each result.

The ice crystal analogues were produced using a Form 2 3D printer (Formlabs), which achieves a high level of precision

with a minimum layer thickness of $25\mu$m and a laser spot size of $140\mu$m. The maximum dimensions of particles ranged from

1 to 3 cm, with aspect ratios varying between 0.04 and 0.2, and area ratios varying between 0.2 and 1 (Table 1). Due to an

artefact of how the numerical code from Reiter (2005) was used to create some of the crystal shapes, a few of the models (S,



**Table 1.** Particle shapes analysed in this study. Area ratio is calculated using the observed projected area divided by the circumscribing circle around the maximum diameter, as seen from beneath when fall motion is steady. Re is calculated using the observed mean velocity and maximum observed diameter as seen from beneath.

| Shape | Abbreviation | Image | Area ratio | Aspect ratio | Re Range |
|---|---|---|---|---|---|
| Circular disc | CD | | 1.0 | 0.04, 0.1, 0.2 | 3 - 1660 |
| Hexagonal plate | HP | | 0.87 | 0.04, 0.1, 0.2 | 7 - 1680 |
| Wang Sector plate | Wang-S | | 0.80 | 0.025 | 9 - 1567 |
| Broad-branched plate | BBP | | 0.64 | 0.04, 0.07, 0.1 | 5 - 1104 |
| Plate-branched | PB | | 0.56 | 0.04, 0.07, 0.1 | 23 - 1675 |
| Wang Broad-branched plate | Wang-BBP | | 0.5 | 0.025 | 6 - 1542 |
| Fernlike dendrite | F | | 0.47 | 0.04, 0.07, 0.1 | 21 - 1831 |
| Dendrite-V1 | D1 | | 0.39 | 0.04, 0.07, 0.1 | 10 - 1615 |
| Dendrite-around-plate | DP | | 0.34 | 0.04, 0.07, 0.1 | 15 - 1811 |
| Dendrite | D | | 0.31 | 0.04, 0.07, 0.1 | 17 - 2007 |
| Stellar dendrite | S | | 0.23 | 0.04, 0.07, 0.1 | 15 - 2162 |

F, D, DP, and PB) were later realised to be non-hexagonally symmetric and instead have a horizontal aspect ratio (the diameter in the a-axis to the diameter of the a'-axis) of 1, instead of 1:1.15 for a regular hexagon. We do not expect this to affect the broad behaviour of their fall motions, and indeed we observe zigzag, spiral, and transitional behaviour for these particles, but

as noted later, this asymmetry may influence the details of the critical axis that zigzag motions are oriented around.

To replicate atmospheric conditions in the laboratory, the dynamical similarity experiment was conducted in a transparent acrylic tank with internal dimensions of 0.4×0.4×1.8m. The tank was filled with uniform mixtures of water and glycerol, with the volume fraction of glycerol ranging from 0% to approximately 50%. By varying both the density and viscosity of the fluid, and the size of the analogues, it was possible to sample a wide range of Reynolds numbers for each shape (Table 1).

During the experiment, the ice particle analogues were allowed to free-fall through the tank and recorded using three orthogonal cameras. Each camera records the fall of the particle through a region approximately 0.2×0.2×0.2m in size, 1.5 m below the surface of the fluid. By this point, the particles have reached their terminal velocities and their behaviour is insensitive to the initial release orientation.

The Trajectory Reconstruction Algorithm implemented through Image anaLysis (TRAIL) then produced digital reconstruc-
tions of the trajectory and orientation of the particle in free fall. More details on the fabrication of the analogues, experimental setup, and reconstruction algorithm can be found in McCorquodale and Westbrook (2021a).




This data, referred to as TRAIL, provides time series of the 3D positions and orientation of the falling analogues, from which the 3D velocity vectors at each time step can be derived. A total of 354 experiments with plate-like shapes were conducted, resulting in the range of values described in Table 1.

The instantaneous velocity at each time step is calculated by applying the central difference formula to the coordinate values, providing an estimate of the instantaneous velocity of the particle at each time point.

## 3   Results

Motion observed in the laboratory was typically stable or periodic. Based on the variation of the particle inclination angle, $\theta$ (Figure 1), the periodic behaviour can be divided into 3 sub-types: zigzag, spiral, and transitional behaviour, and will be analysed below.

### 3.1   Crystals which fall steadily

Stable motion corresponds to a constant, near-zero inclination angle, which is defined in TRAIL as the Euler angles of the particle remaining below a $\pm 2.5°$ threshold value, corresponding to the resolution of the 3D reconstruction. Stable particles fall horizontally: with their a-axis in the horizontal plane and c axis oriented vertically, with no measurable fluctuations in velocity, and no horizontal movements.

223 ice crystal analogues exhibited stable motion, while 131 exhibited unstable, periodic motion. Across all shapes, Reynolds number alone cannot be used to predict stability: the Reynolds numbers observed ranged from 3 – 1615 for stable motion and 197–2162 for unstable motion. The range of moment of inertia values was $0.14 \cdot 10^{-3}$–$12 \cdot 10^{-3}$ for stable motion and $0.28 \cdot 10^{-3}$–$11 \cdot 10^{-3}$ for unstable motion. With both variables exhibiting a considerable overlap in presented behaviours, the onset of stability for ice crystals cannot be considered the same as for circular discs, which become unsteady around Re = 100-200 (Field et al., 1997) and around Re = 200 for our results (Figure 3).

Shape (approximated by area ratio) is found to have a large impact on instability. The coverage of stable and unstable behaviours for all ice crystal analogues in TRAIL is summarised in Figure 3, and it can be seen that onset of stability can be at larger Re (by up to an order of magnitude) than the predicted onset of unsteadiness for circular discs. The spread of experiments and their motion types by Reynolds number, separated by shape, is presented in Figure 3.

Some particles are stable for a much larger range of Reynolds numbers than others. A few shapes (D1 at all aspect ratios, as well as DP, S, and F at aspect ratio 0.04) remained stable throughout all conditions, even at $Re > 10^3$. In McCorquodale and Westbrook (2021b), there is evidence that the drag coefficient increases when planar crystals become unsteady. This change in $C_D$ is more pronounced when area ratio is high than when it is low, suggesting that unsteadiness is less vigorous in particles with low area ratios, such as dendrites.





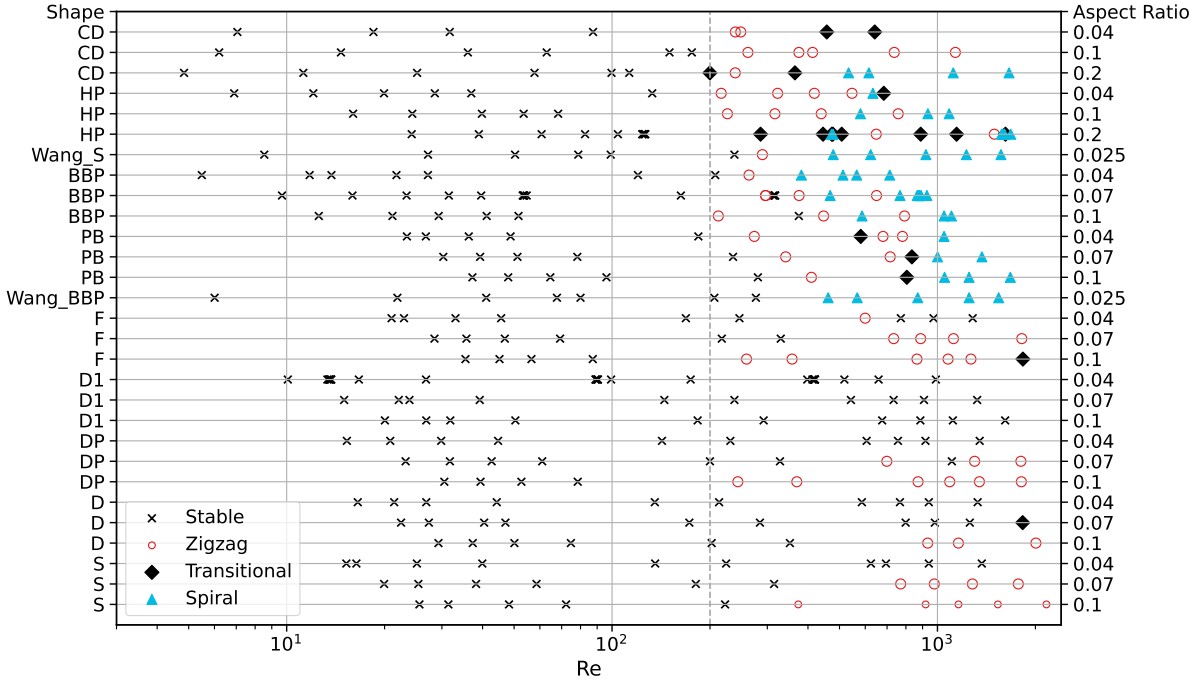

**Figure 3.** Motion type coverage of Reynolds number for each shape and aspect ratio. Stable, zigzag, transitional, and spiral, are black crosses, pink circles, black diamonds, and blue triangles respectively. Particle shape labelling is defined in Table 1

## 3.2 Case studies of periodic motion

Four case studies are presented to illustrate the periodic motion sub-types, seen in Figure 4 and described in Table 3. In this section, we will quantitatively describe the 4 case studies, and then objectively classify their motion based on inclination angle.

Each case study experiment was conducted in pure water. The case study examples are of hexagonal plates except the spiralling case (Fig. 4g,h) which was a broad-branched plate as none of the hexagonal plate studies exhibited pure spiralling behaviour with no wobble, but instead exhibited transitional-spirals. Figure 4a, c, e and g present side-views of the particle cases, viewed from a lab frame of reference. Figure 4b, d, f, h, present the linearly-detrended centre of mass of each particle at each timestep, effectively subtracting the mean fall velocity, such that the particle is viewed from an observer falling at the same mean velocity as the particle.

### 3.2.1 Characteristics of periodic motion

The first of the periodic motion types seen is the zigzag case study: the particle swings back and forth in one plane, and as the particle swings away from its centre of fall, its inclination angle increases, akin to a planar pendulum motion. The zigzag-transitional case introduces an element of rotation around the vertical axis, such that the plane of swing slowly moves anticlockwise, and had the experimental run been longer, it may have rotated back to its original position. The transitional-





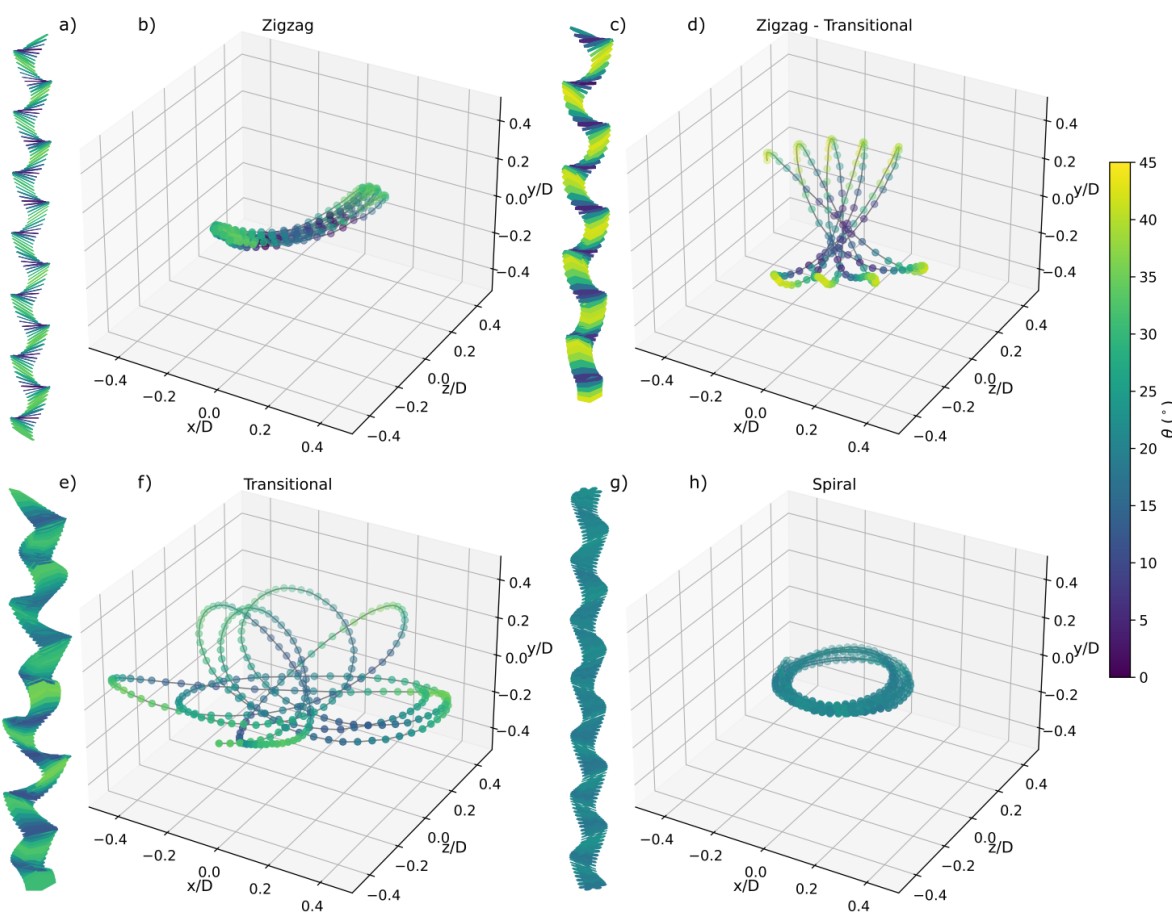

**Figure 4.** Case studies visualising the periodic motion sub-types. Side views of the particle motion (a, c, e, g), and linearly de-trended centre of mass normalised by particle diameter (b, d, f, h), coloured by inclination angle $\theta$ (°) for the Zigzag, Zigzag-Transitional, Transitional, and Spiral cases respectively.

spiral case is similar, but the rate of rotation around the vertical is faster, producing wider loops. Spiralling, the final sub-type of periodic motion remains at a near-constant inclination angle and does not swing back and forth, and instead precesses around its central point without touching its mean centre of fall.

The sub-types of periodic motion can be approximated by the sub-types of spherical pendulums: zigzagging is similar to a planar pendulum, spiralling is comparable to a conical pendulum, and transitional motion captures the range of pendulum

motion between the two extremes, with the horizontal displacement approximating a rhodonea curve (Helt, 2016).



**Table 2.** The experimental conditions for the presented case studies from Fig.4-8.

| Motion type | Shape | Aspect ratio | Reynolds number | $I* \cdot 10^{-3}$ |
|---|---|---|---|---|
| Zigzag | HP | 0.04 | 546 | 1.59 |
| Zigzag-Transitional | HP | 0.10 | 757 | 3.94 |
| Transitional | HP | 0.04 | 684 | 1.58 |
| Spiral | BBP | 0.04 | 512 | 1.07 |

### 3.2.2 Time series of $\theta$ and $\phi$

Series of inclination angles, $\theta$, for the periodic motion types can be approximated as sinusoidal (Figure 5). To distinguish between the regimes, rectified sine waves are fit to these inclination angle time-series, using a fast Fourier transform as a first guess of the frequency of the sine wave, and SciPy's curve fit function (Virtanen et al., 2020), such that:

$$\theta = |\theta_{\mathrm{amp}} \sin(\omega t + \omega_0) + \theta_{\mathrm{tilt}}| \tag{4}$$

Where $\theta$ is the inclination angle, t is the time in seconds, $\theta_{amp}$ is the amplitude of the wave, and $\theta_{tilt}$ is the angular displacement of the sine wave, the period of the sine wave is $2\pi/\omega$ (in seconds), and $\omega_0$ is the phase shift.

$\theta_{amp}$ and $\theta_{tilt}$ are found to summarise the motion types well, as they represent the variability and tilt of the particle respectively and constrain the pendulum model. They also allow the periodic sub-types to be distinguished quantitatively: a spiralling particle has a low $\theta_{amp}$ and a high $\theta_{tilt}$, as it is consistently inclined and does not flutter.

Zigzagging behaviour is the opposite: a potentially high $\theta_{amp}$ and a near-zero $\theta_{tilt}$, as it swings around a horizontal orientation, but flutters much more than a spiralling particle (Fig. 5). For example, the zigzag example case (Fig. 5a) has a fitted $\theta_{amp}$ of 34 °, and a $\theta_{tilt}$ of 0 °.

The transitional zigzag case behaves similarly, but never samples (close to) $\theta = 0$, and it is just starting to transition to having a slow rotational component. It should be noted that although the transitional-zigzag case has a higher $\theta_{amp}$ (42°), than the zigzag case (where $\theta_{amp}$ is 34°), this does not negate categorisation of behaviour for either case, as the trajectory of the transitional-zigzag case is close to zigzag behaviour but never samples exactly $\theta = 0$, and it is just starting to transition to having a slow rotational component.

The transitional case (Fig 5c) has a smaller amplitude than both zigzag and zigzag-transitional cases; $\theta_{amp}$ is 9°, but $\theta_{tilt}$ is much higher, at 24°. The spiralling case (Fig 5d) has an even smaller amplitude still, with $\theta_{amp}$ is 1°, but $\theta_{tilt}$ at 20°. Spiralling behaviour can have nonzero $\theta_{amp}$, although $\theta_{amp}$ is small. This small variation in $\theta$ is referred to as wobble for spiralling cases. It may be worth considering how much of this wobble is a physical phenomenon vs an artifact or experimental uncertainty; a wobble of $2.5^o$ could easily originate from experimental uncertainty. Given the wobble in figure 5d appears to have a uniform period, we believe the wobble in this case is partly a physical phenomenon, but you can see the impact of experimental uncertainty in the traces within figures 5(a-c) at the limits of inclination angle (e.g. for zigzag the angle often




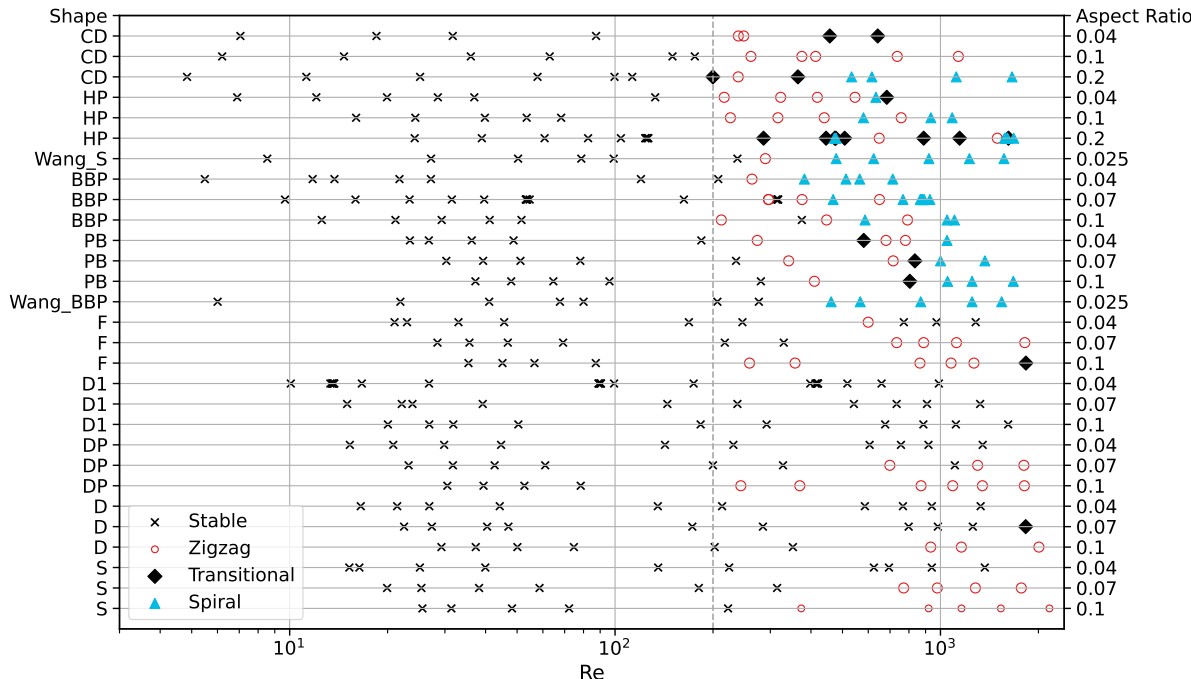

**Figure 5.** Time-series of inclination angles (°) for the periodic motion sub-types shown in Figure 4 A histogram of angles is shown to the right of each panel, using 2.5 ° bins.

doesn't reach 0). This will partly be due to the finite time-resolution of the measurements and partly due to accuracy of the orientation reconstruction.

Figure 5 also displays the distributions of $\theta$ for each case study. The zigzag case has a distribution that is consistent with that expected for simple harmonic motion, where the most likely inclination is at the end of each swing where $d\theta/dt$ is smallest,
305 and least likely is an angle of zero. Spiral has an almost constant inclination angle, and hence a very narrow distribution of $\theta$, centred on angle significantly higher than zero. Transitional has a distribution that is in-between the other two cases, and in common with spiral cases $\theta$ is always above zero. This is in significant disagreement with the common assumption that orientation is a Gaussian distribution where most common orientation is horizontal ($\theta = 0$).

An azimuth angle, $\phi$, represents where the c-axis of the crystal when projected into plan view is pointing relative to the x
310 axis (in the lab reference frame) (Figure 1c). A spiralling particle has a linear increase (or decrease, in cases not shown) of azimuth angle, and $d\phi/dt$ is constant such that it precesses at a constant rate. The saw-tooth shape is produced by the angle being limited to $\pm 180°$.

Purely zigzag cases see-saw around one axis: when the particle goes from pointing one way to pointing another, $\phi$ changes by $\Delta 180°$ seen by the square-wave shape. $d\phi/dt$ is constant and zero for pure zigzag cases except for close to the instant
315 where the particle becomes horizontal ($\theta = 0$) and $\phi$ becomes highly uncertain, as the c-axis is momentarily pointing towards



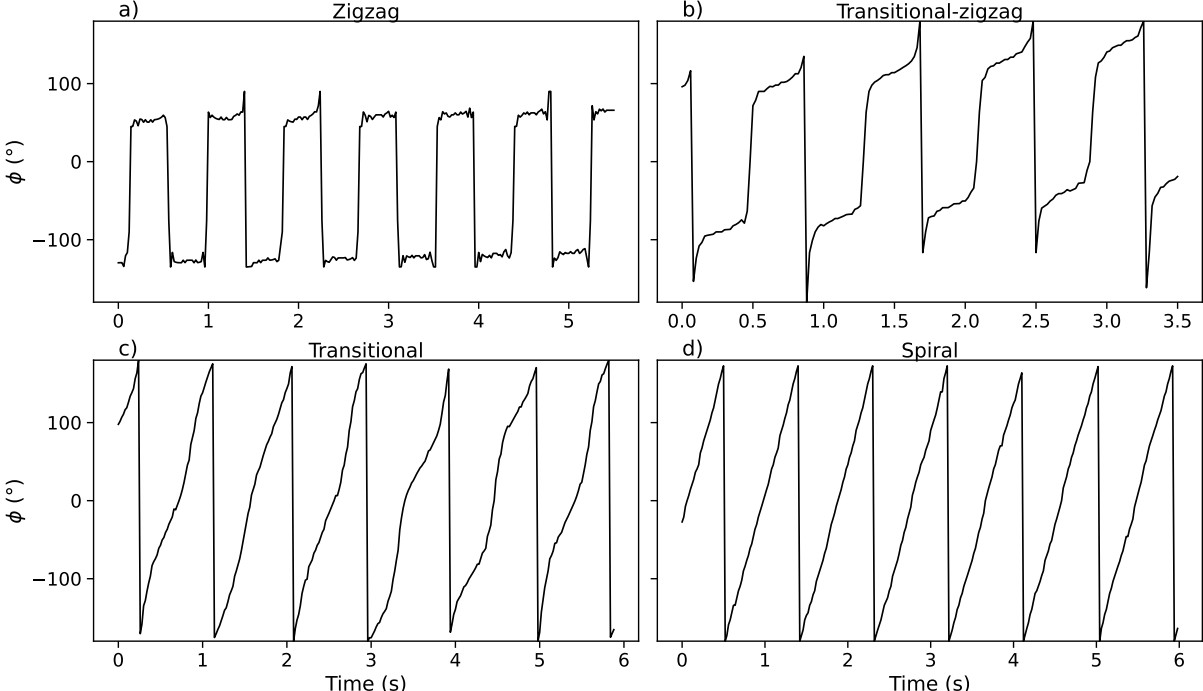

**Figure 6.** Time-series of azimuth angles, $\phi$, (°) for the periodic motion sub-types.

the vertical (and can be ignored). The axis that zigzagging cases see-saw around tends to be the arms of the crystal, in the plane of the a and a'-axes. For shapes that are non-hexagonally symmetric (S, F, D, DP, and PB), the shortest arms are the axis the crystal see-saws around.

Transitional cases have a non-constant $d\phi/dt$, a combination of the saw-tooth and square waves seen in the spiral and zigzag cases. For the zigzag-transitional case, $\phi$ increases during its time along each loop of the rhodonea curve, and then jumps by a value close to $180°$ when the orientation of the c-axis is close to vertical. The transitional case has no visible jumps in $\phi$ except for the aliasing at $\pm 180°$, as $\theta$ never becomes close to zero.

### 3.2.3 Velocity fluctuations

The amplitudes of the u and w components of velocity (in the x and z directions respectively) changed in the presence of any component of rotation around the vertical. Therefore, a combined horizontal component of velocity, U, was calculated as follows:

$$U = \sqrt{u^2 + w^2} \tag{5}$$



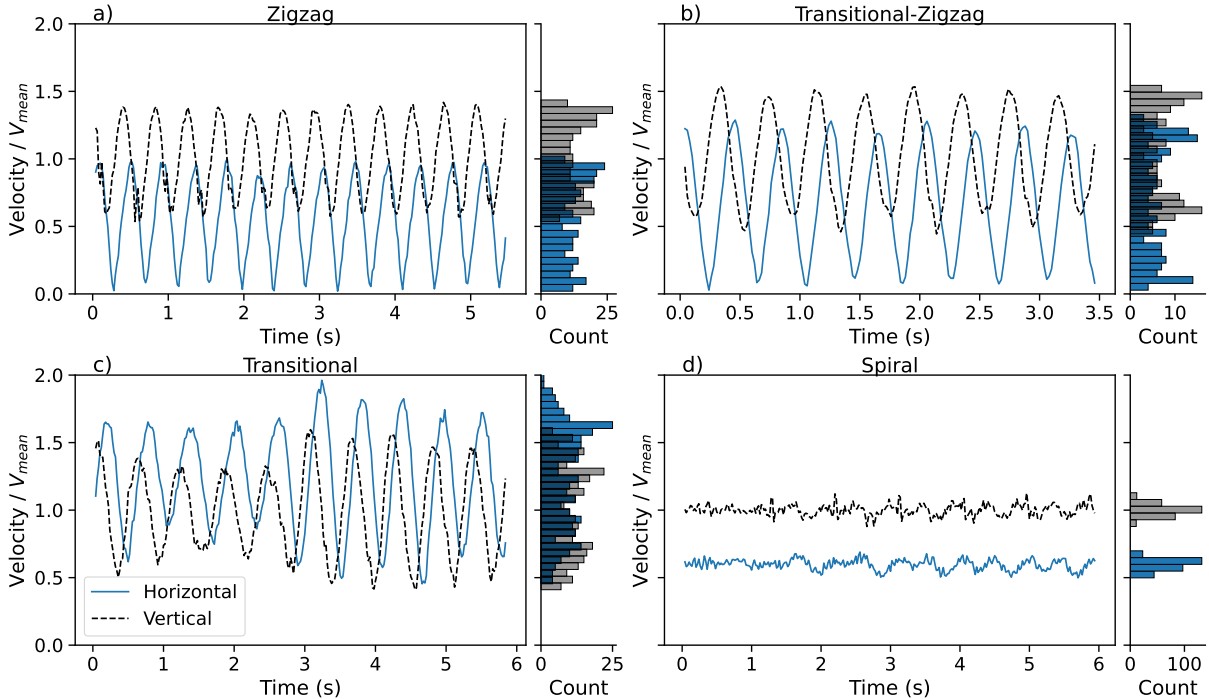

**Figure 7.** Time-series of the horizontal and vertical velocity components for the periodic motion sub-types, normalised by the mean vertical velocity in each experiment.

Sine waves can be fit to the vertical component of velocity, V, and the horizontal component of velocity, U, such that:

$$V = V_{\mathrm{amp}}\sin(\omega_V t + \omega_{V0}) + V_{\mathrm{offset}} \qquad (6)$$

$$U = |U_{\mathrm{amp}}\sin(\omega_U t + \omega_{U0}) + U_{\mathrm{offset}}| \qquad (7)$$

These velocity components of the particles were found to follow a sinusoidal pattern consistent with the pendulum model (Fig. 7). $U$ is fit with a rectified sine wave as horizontal speed can become zero in zigzag cases, but cannot be negative (equation 5).

For all experiments with periodic motion, $\theta$, $U$, and $V$, are all found to have sinusoidal patterns with the same period, (i.e. $\omega_V = \omega_U$ ) but different offsets and amplitudes. $U_{\mathrm{offset}}$ is not always zero: it is non-zero for particles that drift as they descend, or particles with a spiralling component.

A summary of the sine wave fit components can be found in Table 3. Horizontal velocity, U, is found to peak when the tilt was lowest (i.e., when the particle was flat or the angle of the particle was closest to zero) just after vertical velocity, V, peaks.

The amplitude of V fluctuations relative to the mean vertical fall speed ($V_{amp}/V_{mean}$) are 0.37, 0.47, 0.42, and 0.03 for the zigzag, zigzag-transitional, transitional, and spiral cases respectively. The amplitude of horizontal velocity fluctuations relative to the mean fall speed ($U_{amp}/V_{mean}$) are similar to their vertical components (0.41, 0.55, 0.49, 0.03 for each case respectively).




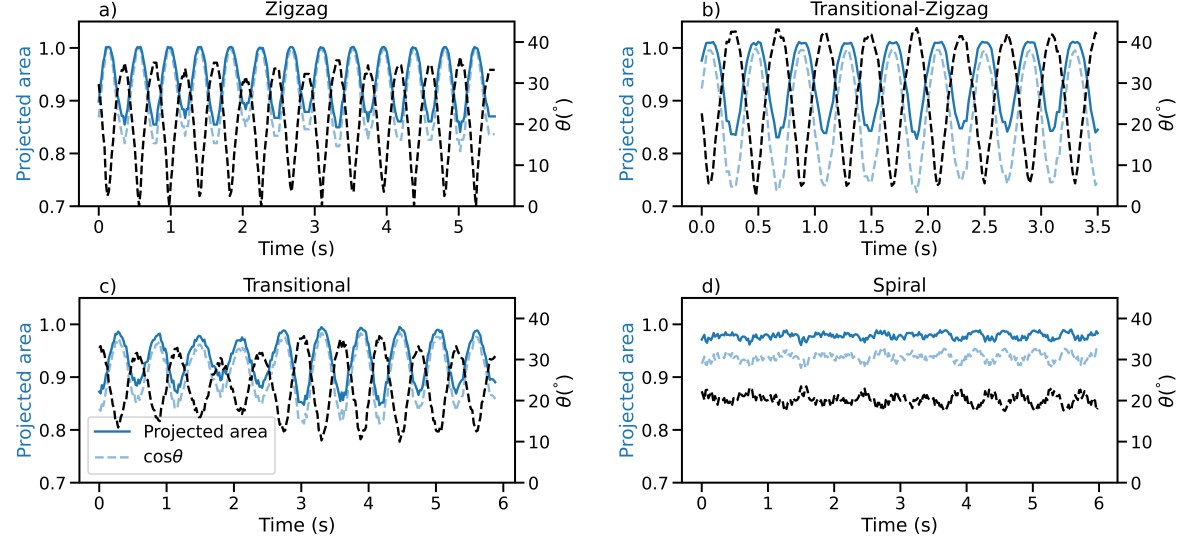

**Figure 8.** Time-series of modelled projected areas as seen from below for each case study, normalised by the maximum observed projected area (blue). $\theta$ (dashed, black) and $cos\theta$ (dashed, light blue) provided for reference.

In the case of the spiralling particle, U and V are held relatively constant compared to the other cases, effectively making the spiralling cases a quasi-steady mode with a non-zero near-constant inclination.

For the non-spiralling cases, large fluctuations suggest that the mean vertical velocity, $V_{mean}$, does not sufficiently characterise the velocity of a particle. The distributions shown in Figure 7 for the non-spiralling cases are broad, suggesting that a broad spectrum of instantaneous velocities for a single type of oscillating particle should be considered when interpreting Doppler spectra.

### 3.2.4 Projected area fluctuations

One major application of this research is for dwelling radars and lidars, whether ground-based (usually close to zenith) or spaceborne (usually close to nadir). Variation in projected area affects assumptions in backscatter cross-section and hence retrieval of particle size and number. Fluctuations will also affect polarimetric measurements and retrievals using these, particularly when the particles are viewed from the side. Figure 8 presents the timeseries of projected areas as seen from below for each case study, normalised by the planar cross sectional area of each analogue. Projected area as seen from below anti-correlates with $\theta$ in each time-series, displaying an out-of-phase relationship: as $\theta$ increases, the aspect ratio of the analogues increases (becoming closer to 1), while the area ratio and projected area decrease.

For all four cases, vertical velocity and projected area exhibit a $180°$ out-of-phase relationship, where vertical velocity peaks just after projected area reaches its minimum point. Projected area is more in-phase with horizontal velocity, peaking just after




$U$ reaches its maxima. This supports the idea that vertical velocity increases when projected area is minimised, as drag is minimised in the horizontal plane, allowing the particle to accelerate.

In the case of an infinitely thin particle, projected area as seen from below is equal to the cross-sectional area of the particle multiplied by $cos\theta$ (and hence correlated with $cos\theta$, as evident from Figure 7). In the presence of particle thickness, the normalised projected area is expected to be greater than or equal to $cos\theta$, and therefore should never fall below 0.7, as $\theta$ never exceeds $45°$ for periodically oscillating cases.

### 3.2.5   Motion type parameter, $\epsilon$

Since these four cases are not discrete classes of behaviour, we propose a parameter to characterise where each experiment lies on the continuum of motions between zigzag and spiral. To quantify the spectrum of periodic behaviour, a motion type parameter, $\epsilon$, is defined as:

$$\epsilon = \frac{\theta_{tilt}}{\theta_{tilt} + \theta_{amp}} \tag{8}$$

Such that particles with $\epsilon = 1$ correspond to spiralling, as $\theta_{tilt} \gg 0$, and $\theta_{amp} \approx 0$ for spiralling cases. Zigzagging behaviour
corresponds to $\epsilon = 0$, as zigzagging behaviour has high amplitudes, $\theta_{amp} \gg 0$, and $\theta_{tilt} \approx 0$. Transitional cases can have nonzero $\theta_{tilt}$ and $\theta_{amp}$, and $\epsilon$ can therefore range between 0 and 1. The four case studies presented have $\epsilon = 0.01, 0.00, 0.73$, and 0.94 for zigzagging, zigzag-transitional, transitional, and spiral respectively.

### 3.2.6   Oscillation frequencies

For bulk approximations (retrievals, microphysics schemes), it is useful to characterise the frequency of oscillatory behaviour.
To nondimensionalise the frequency of oscillation of the experiments, Strouhal number is calculated as:

$$St_\theta = \frac{fD}{v} \tag{9}$$

where f is the frequency of oscillation found by the sine waves fit to $\theta$ (Kajikawa, 1992). $St_\theta$ is therefore representative of the number of oscillations in $\theta$ of the particle in the time it takes for the particle to fall the vertical distance equal to its diameter.

    $St_\theta$ (frequencies) of the four cases are 0.45, 0.34, 0.91, and 0.90 for zigzag, zigzag-transitional, transitional, and spiral
respectively (Table 3).

    A secondary Strouhal number can also be calculated using the rate of change of $\phi$:

$$St_\phi = \frac{1}{360°} \left| \overline{\frac{d\phi}{dt}} \right| \frac{D}{v} \tag{10}$$

where $\left| \overline{\frac{d\phi}{dt}} \right|$ is the mean absolute rate of precession of the particle. $St_\phi$ is therefore the number of full turns around a vertical axis that a particle makes during the time it takes for the particle to fall the vertical distance equal to its diameter. $St_\phi$ for the
zigzag and zigzag-transitional cases are 0.028 and 0.079 respectively. The zigzag case is effectively nonrotational around the vertical axis, and therefore has the lowest $St_\phi$. The transitional and spiral cases have substantial rotation around the vertical





**Table 3.** The observed behaviours of the presented case studies.

| Motion type | $\theta_{mean}$ (°) | $\theta_{tilt}$ (°) | $\theta_{amp}$ (°) | $\epsilon$ | $V_{amp}/V_{mean}$ | $U_{amp}/V_{mean}$ | $St_\theta$ | $St_\phi$ |
|---|---|---|---|---|---|---|---|---|
| Zigzag | 22 | 0 | 34 | 0.01 | 0.37 | 0.41 | 0.45 | 0.028 |
| Zigzag-Transitional | 27 | 0 | 42 | 0.00 | 0.47 | 0.55 | 0.34 | 0.079 |
| Transitional | 24 | 24 | 9 | 0.73 | 0.42 | 0.49 | 0.91 | 0.57 |
| Spiral | 20 | 20 | 1 | 0.94 | 0.03 | 0.03 | 0.90 | 0.46 |

axis, and therefore have $St_\phi$ of 0.57, and 0.46 respectively. The transitional case spirals faster and wobbles faster than the spiral case, but otherwise both $St$ and $St_\phi$ appear to increase with $\epsilon$.

### 3.3 Characteristics of the full data set

#### 3.3.1 Distributions of $\theta$

Figure 9 displays the distributions of inclination angle across all Re, excluding stable experiments, and separates in particular to demonstrate its impact on the distributions. Onset of unsteadiness is seen at Re ≈ 200, (Re = 212 for non-circular disc shapes) and lowest onset of spiralling in particular is seen at Re = 461 (for non-circular disc shapes), and the onset of spiralling for circular discs is seen as low as Re ≈ 300.

Different shapes show different distributions of inclination angles, as well as exhibiting different $\theta_{tilt}$ and $\theta_{amp}$ (Figure 9). For a particular shape, the distribution remains similar between adjacent Reynolds number bins; once a specific motion regime is reached for a particular shape, the same $\theta_{tilt}$ and $\theta_{amp}$ are maintained. For instance, Wang-BBP shapes spiral and have a mean $\theta$ of around 21 degrees (along with a narrow distribution), while circular discs tend to have a much wider distribution, corresponding with high amplitude zigzag behaviour.

Many of the shapes that present spiralling behaviour at high Re first present zigzagging behaviour at intermediate Re (see also Figs. 3 ,11, and 10).

#### 3.3.2 Characterisation of motion

Sine waves were fit using equation 7 to all velocity components and to $\theta$ for all periodic experiments (equation 4), and the motion type parameter $\epsilon$ was subsequently calculated (Figure 12) (equation 8).

There is discrepancy between the literature on circular discs and our observations of ice crystal shapes when taking I* into account. The motion parameter, $\epsilon$, increases for increasing Reynolds number and dimensionless moment of inertia (Fig. 10, which is the opposite of the result found in Zhong et al. (2011), who found that, for circular discs exhibiting periodic motions, spiralling occurred at lower Re and I*, and zigzagging occurred at higher Re and I*. Our result instead agrees with Cheng




**Figure 9.** Inclination angle distributions of all unstable experiments, equally weighted by experiment, binned by Reynolds number and particle shape, with shapes in order of decreasing area ratio, excluding CD-P and D1. Experiments are split by spiral, $\epsilon > 0.5$ (upper, orange) or zigzag ($\epsilon \leq 0.5$) (lower, blue). Quartiles (dashed) and mean values (solid) are inside each distribution. $\overline{\theta}_{amp}$ $\overline{\theta}_{tilt}$ are the mean values of $\theta_{amp}$ and $\theta_{tilt}$ for each set of experiments (separated by $\epsilon$ and particle shape).





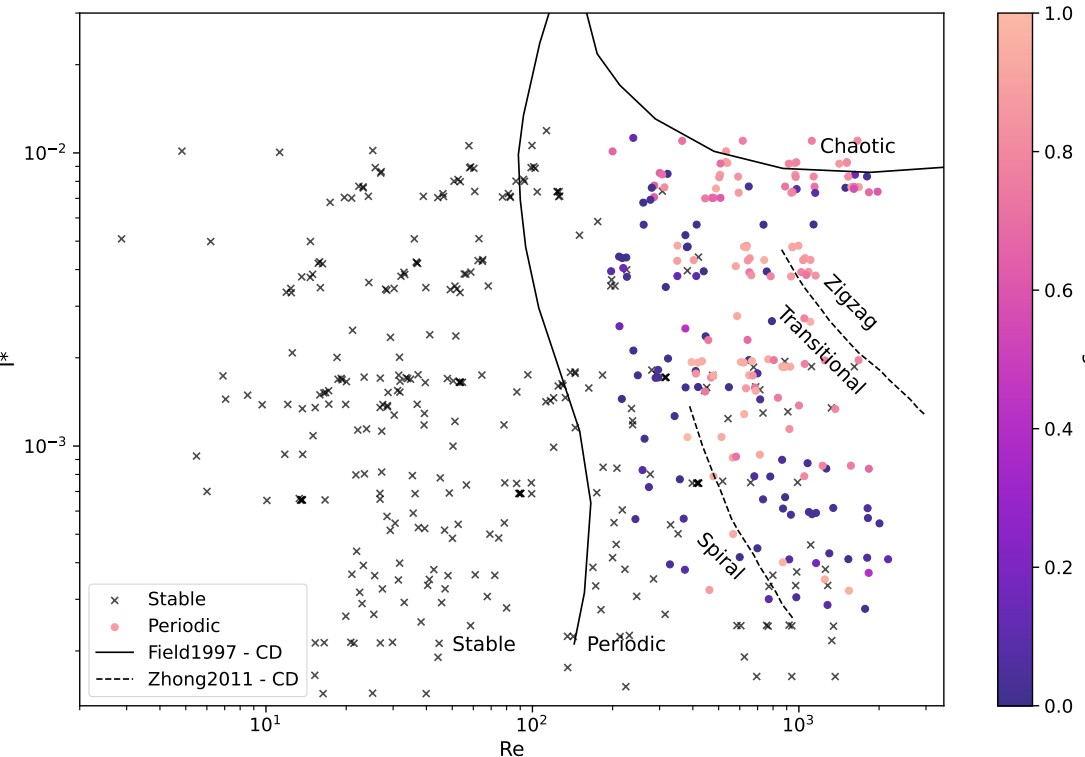

**Figure 10.** As in Figure 2, but with each experiment in TRAIL coloured by $\epsilon$. stable experiments are marked with black crosses. Solid lines are from Field et al. (1997) and dashed lines are from Zhong et al. (2013).

et al. (2015), who found that hexagonal plates exhibit a zigzag motion at low Re while larger plates at higher Re exhibited
spiralling. Jayaweera (1965) also finds that, for falling spheres, zigzagging occurs at lower Re than spiralling, which occurs at
very high Re ($Re > 10^5$). Our results also deviate from the stable-periodic division line provided by Field et al. (1997), as the
ice crystal shapes do not become unsteady until higher Reynolds numbers than circular discs, as shape has a strong impact on
the conditions that the onset of unsteady motions occur.



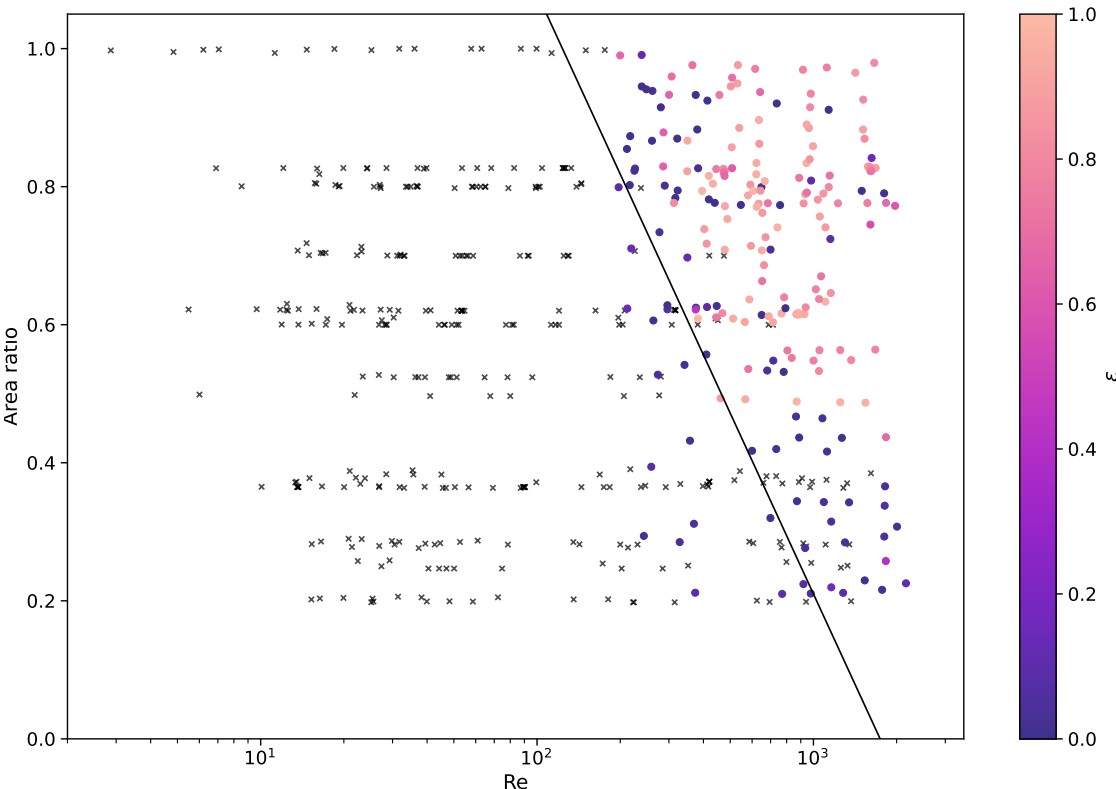

**Figure 11.** Phase diagram showing the stable (black crosses) and periodic (coloured by $\epsilon$) behaviour of falling particles as a function of Area ratio and Reynolds number. Equation 11 is shown as the black line.

Whilst I* is used successfully for studies on circular discs, shape must also be taken into account when considering the broad range of shapes that ice crystals exhibit. Differences in shape can be quantified by area ratio (the ratio of the maximum cross-sectional area of the particle and the area of its circumscribing circle).

Across the experiments, $\epsilon$ increases for increasing Reynolds number and area ratio (Figure 11. In agreement with Esteban et al. (2019) and Tagliavini et al. (2021), stable fall behaviour is found to be much more likely for particles with lower area ratios — i.e., ice crystals with more dendritic or complex shapes are more likely to fall steadily when under the same Reynolds number.



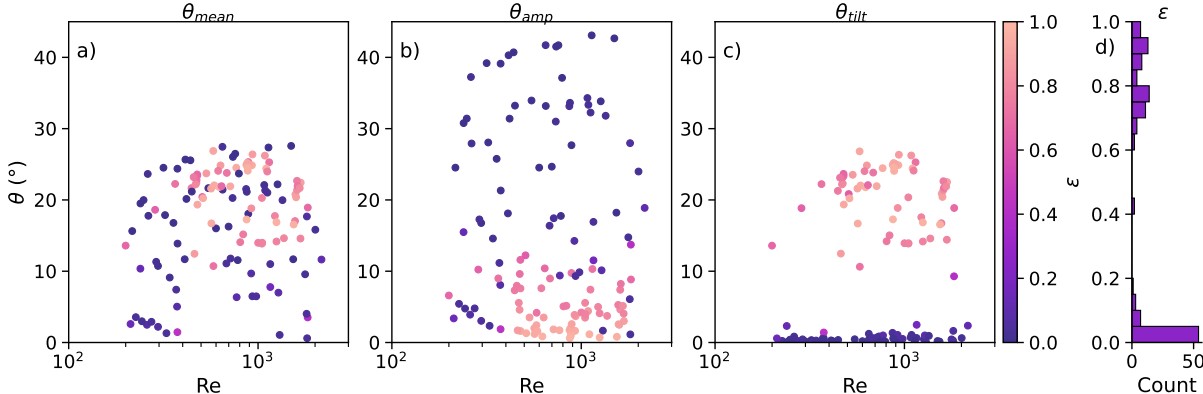

**Figure 12.** Mean inclination angle, $\theta_{mean}$ (a), $\theta_{tilt}$ (b) and $\theta_{amp}$ (c) by Reynolds number, coloured by $\epsilon$. Histogram of $\epsilon$ (d).

Using linear support vector classification (Pedregosa et al., 2011), we identify a line of best fit that maximises the distance between stable and periodic behaviour, such that periodic behaviour occurs when:

$$\log_{10}(Re) > \frac{2.82 - Area\ ratio}{0.87} \tag{11}$$

The critical point of this expression is displayed in Figure 11.

Mean inclination angle was not found to distinguish well between periodic motion types (Fig. 12a). $\theta_{tilt}$ is found to be near-zero for zigzagging cases (where $\epsilon < 0.5$), but $\theta_{amp}$ can reach up to $45°$(Fig. 12b). For all experiments where $\epsilon > 0.5$, $\theta_{tilt}$ is between 7 and 28 degrees (Fig. 12c).



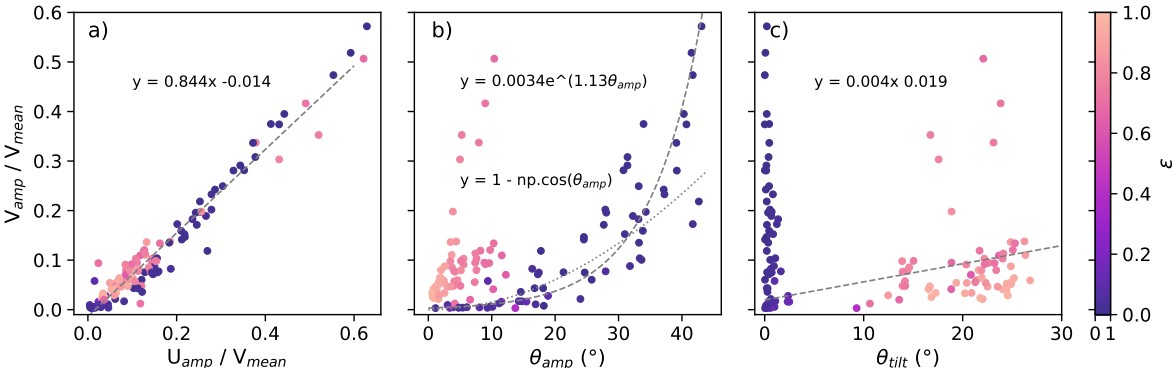

**Figure 13.** Variation of components of motion for all periodic experiments. Amplitude of horizontal velocity relative to mean vertical velocity compared to: amplitude of vertical velocity relative to mean vertical velocity (a), $\theta_{amp}$ (b), $\theta_{tilt}$ (c), coloured by $\epsilon$. Dashed lines are fit are fit to all experiments (a), experiments where $\epsilon$ <0.5 (b) and experiments where $\epsilon$ >0.5 (c)

The distribution of $\epsilon$ was found to be bimodal (Figure 12d) favouring either zigzag or spiralling behaviour, with transitional motion being less likely. Out of all 131 periodic platelike observed experiments, 65 experiments displayed $\epsilon < 0.2$, 34 were found to have $\epsilon$ between 0.2 and 0.8, and 32 experiments were $\epsilon > 0.8$. The potential cause of this is discussed in Section 3.3.4.

### 3.3.3  Velocity fluctuations

Across all periodic experiments, the amplitude of the vertical velocity, $V_{amp}$, was found to be approximately 85% of the amplitude of the horizontal speed, $U_{amp}$ (Fig. 13a), using least-squares linear regression. In contrast to our findings, Kajikawa (1992) reported that the standard deviation of the horizontal velocity, $U$, was considerably larger (5 to 20% of the fall velocity) than the standard deviation of the vertical velocity (<3% of the fall velocity) for dendritic-shaped particles undergoing periodic oscillation. The reason for the difference between our findings and those of Kajikawa is unknown and it is hard to understand why fluttering particles would have large horizontal velocity fluctuations but almost constant vertical velocity. More investigation of natural particles using modern observations, such as Maahn et al. (2023), may help explore this in the future.

For zigzagging particles, as $\theta_{amp}$ increases, the amplitudes of both the vertical and horizontal speed components increased exponentially (Fig. 13b). For spiralling particles, as $\theta_{tilt}$ increases, amplitude of vertical velocity increases slightly (i.e. there is more wobble). $\theta_{tilt}$ has no influence on amplitude of vertical velocity for zigzagging particles, as it is near-zero (Fig. 13c).

### 3.3.4  Strouhal numbers

Strouhal numbers were calculated for all experiments as detailed in equations 9 and 10. For experiments where $\epsilon \leq 0.2$ $St_\theta$ has a mean of 0.29 and a standard deviation of 0.10, with no significant trend with Re (Figure 14). Zigzagging particles ($\epsilon \leq 0.2$) never have $St_\theta$ above 0.50. Despite having typically much smaller amplitudes than zigzagging particles, spiralling experiments ($\epsilon \geq 0.8$) have a larger range of $St_\theta$, with a potential for $St_\theta$ up to 1.50 and mean $St_\theta$ of 0.5. $St_\theta$ for spiralling particles may be





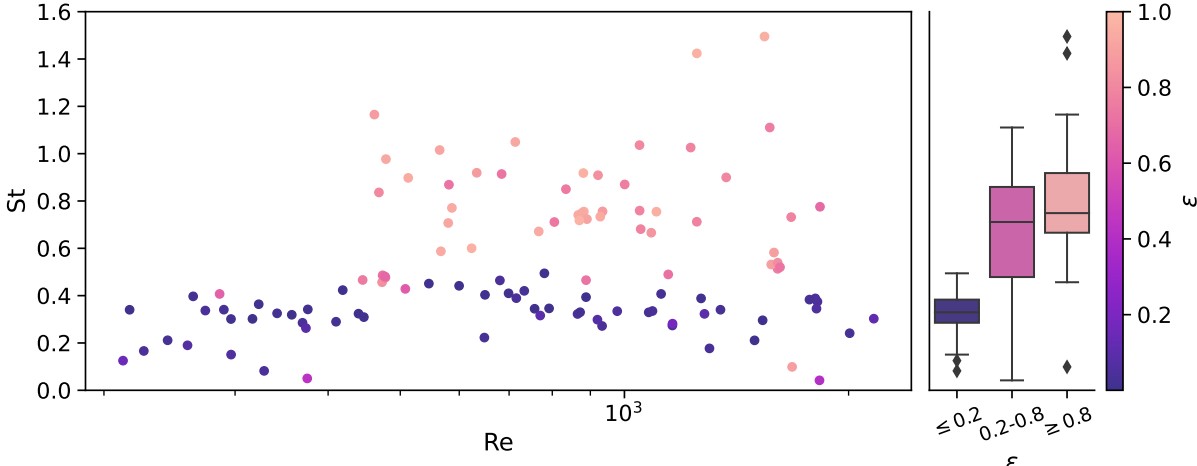

**Figure 14.** Scatter plot of Strouhal numbers ($St_\theta$) for each unstable experiment versus Reynolds number, coloured by $\epsilon$. Box plots of observed Strouhal numbers for particles with $\theta_{amp} > 2.5°$ alongside.

greater than for zigzagging; the wobbling of a spiralling particle is at a much smaller amplitude ($\theta_{amp}$) than that of a zigzagging particle.



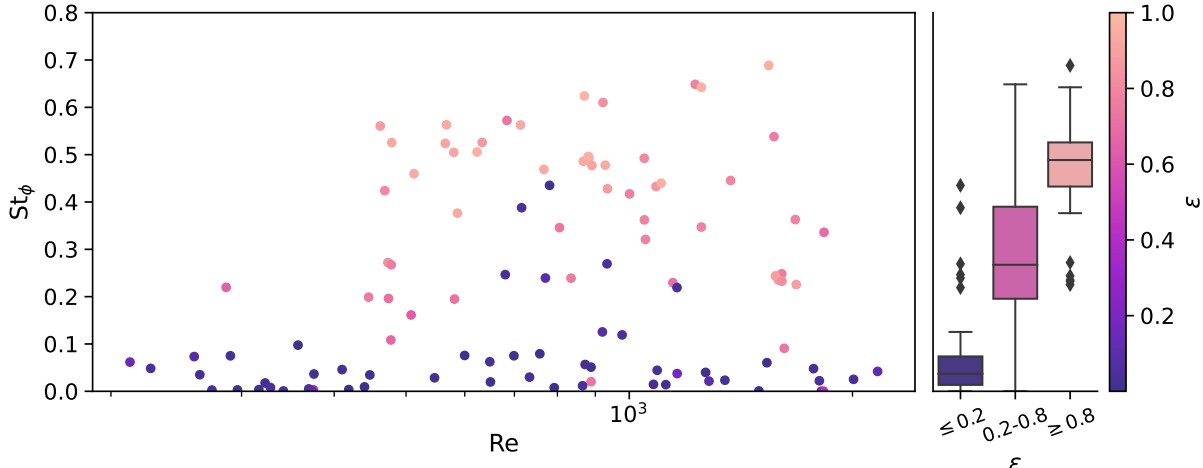

**Figure 15.** Scatter plot of azimuth Strouhal numbers ($St_\phi$) for each unstable experiment versus Reynolds number, coloured by $\epsilon$. Experiments where $\theta_{amp} < 2.5°$ marked with crosses. Box plots of observed Strouhal numbers for particles with $\theta_{amp} > 2.5°$ alongside.

Strouhal number of the vortex shedding frequency of cylinders in flow is a function of Reynolds number, $St$ for a cylinder
increases from $St \approx 0.1$ at $Re \approx 50$ to $St \approx 0.22$ at $Re \approx 2000$, and $St \approx 0.2$ from $Re$ $10^4 - 10^6$ (Katopodes, 2019). $St_\theta$ for the ice analogues may not correspond exactly to the nondimensionalised frequency of vortex shedding but was overall found to be higher than previous work on discs, suggesting that vortices may shed more frequently for more complex shapes.

Similarly to $St_\theta$ and previous literature on circular discs, there is no particular trend in $St_\phi$ with Reynolds number (Figure
15). $St_\phi$ is close to zero for zigzagging particles (where $\epsilon \leq 0.2$), as the rate of spiralling is very low (by definition), whereas
for spiralling particles ($\epsilon \geq 0.8$) , $St_\phi$ can be as high as 0.7.



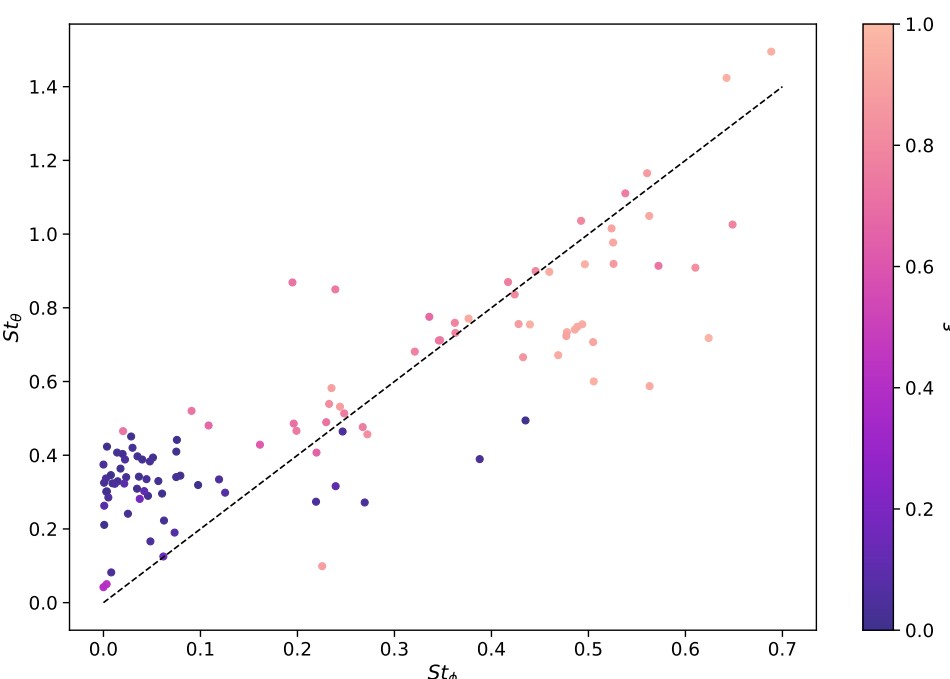

**Figure 16.** Scatter plot of azimuth Strouhal numbers ($St_\phi$) vs inclination Strouhal numbers ($St_\theta$). The slope of the dashed line is 2.



When spiralling analogues rotate faster, they tend to also wobble more frequently. For high $\epsilon$ (and correspondingly, small $\theta_{amp}$) $St_\theta$ is approximately half of $St_\phi$, corresponding with the classic observation that wobbling plates are found to wobble twice as fast as they rotate (Figure 16 ) (Tuleja et al., 2007). When $\theta_{amp}$ is nonzero, the spiral motion that the centre of mass of the analogue makes is not a perfect circle: the smaller the angle of wobble, the closer the traces of the analogue are to circles.

When the spin is not a perfect circle $St_\phi$ no longer matches double the wobble rate, $St_\theta$. For higher $\theta_{amp}$ (low $\epsilon$) cases, $St_\phi$ and $St_\theta$ appear to both remain low and not depend on one another.

The higher $\theta_{amp}$ is, the more likely the behaviour is to be transitional (non-perfect spiralling), and the larger the swing the particle makes, and therefore the wobble is less frequent, and $St_\phi$ is lower. This may also be the cause of the bimodal distribution in $\epsilon$ (12). Particles that have high $St_\phi$ spiral quickly relative to their vertical velocity. This fast rotation around

the vertical means that the any torque at $90°$ to the vertical axis of rotation (which causes wobble in $\theta$) will have less effect, because it is small relative to the gyroscopic torque. Therefore, the rotation of spiralling particles likely inhibits any potential zigzagging motion.

## 4    Discussion and Conclusion

Ten different platelike snowflake shapes, in addition to circular discs, of up to three different aspect ratios each were allowed

to free fall through a tank of water-glycerine mixture to simulate behaviours of real ice crystals in the atmosphere. The fall behaviour of these analogues was viewed by three orthogonal cameras, allowing for the digital reconstruction of their trajectories and orientations.

### 4.1    Particle fall motion

Four main falling regimes are observed: stable, zigzag, transitional, and spiralling. Stable motion has no measurable fluctua-

tions, while other regimes involve periodic oscillations in both inclination angle and velocities. All unstable motions for the experimental series observed were of periodic behaviour: no tumbling behaviour is observed in this work. Stable analogues all had theta = 0 - i.e. their maximum dimension was in the horizontal plane. Zigzag motion involves swinging back and forth, while spiralling remains inclined at a constant, nonzero inclination angle.

Spiralling particles rotate steadily around the vertical axis, while zigzagging particles maintain a constant azimuth angle,

$\phi$, that has a square wave (such that the minima and maxima are spaced $180°$ apart). Transitional cases are a mix of the two behaviours: they rock back and forth but also rotate as they do so. Particle components of velocity ($V$ and $U$ for vertical and horizontal respectively) were also found to be sinusoidal with respect to time. Sine waves are fit to time-series of $\theta$ and $U$ and $V$, and the rate of spiralling, $d\phi/dt$, is found through linear regression. The amplitude of the sine wave, $\theta_{amp}$, was found to vary between $0°$ and $43.1°$. Periodic motion is found to be analogous to range of spherical pendulum behaviour, corresponding

to simple harmonic motion. Time-series of $\theta$ and velocities for periodic experiments are therefore sinusoidal, and distributions of $\theta$ have a non-zero mode. Results do not support the common assumption of Gaussian orientation distributions with a zero-modal angle during unstable motions: the distributions of $\theta$ have non-Gaussian distributions and have non-zero modes.





In the spiralling regime, components of velocity are held relatively constant compared to the other cases, effectively making the spiralling cases a quasi-steady mode with a non-zero near-constant inclination. When particles spiral, they are consistently

inclined at an angle, observed to typically be between 7 and 28 degrees. The central line of the sine wave fit, $\theta_{tilt}$, is typically between 8 and 25 degrees for spiralling behaviour for all particles, with a mean of 18.4 ° and a standard deviation of 6.8°.

Strouhal numbers (nondimensionalised frequencies) were found using the rate of spiralling, $d\phi/dt$, and the frequency of the sine waves of $\theta$, finding $St_\phi$ and $St_\theta$ respectively. These each represent the number of turns the particle makes, and the number of wobbles the particle makes, in the time taken for the particle to fall the vertical distance equal to its own diameter.

$St_\phi$ was found to be approximately half of $St_\theta$ for spiralling cases: particles were found to wobble twice as often as they made a full rotation around the vertical. For zigzagging cases, $St_\theta$ is found to be 0.29±0.7 with no variation with Reynolds number.

Onset of unstable motions are found to be more likely for higher area ratios, corresponding to less complex shapes (such as pristine hexagonal plates), in agreement with Esteban et al. (2018) and Tagliavini et al. (2021) . The shapes D1 (at all aspect

ratios), DP, S, and F (at aspect ratio 0.04) remained stable throughout all experiments, even at $Re > 10^3$. Onset for circular discs was found to be as low as Re = 197.

Across a set of experiments for a given particle, at low Reynolds numbers, mean inclination angle, $\overline{\theta}$, is close to zero, as particles are stable, and at higher Reynolds number, particles become unstable and $\overline{\theta}$ increases to a steady, nonzero value. To quantify and compare the onset of unstable motions for different particle properties, a logistic curve is fit to the data using a

least-squares method and the Trust Region Reflective algorithm from SciPy's curve fit function (Virtanen et al., 2020), for each particle shape (Figure 17) and area ratio (Figure 17b) as follows:

$$\overline{\theta} = \frac{\overline{\theta}_{unstable}}{1 + e^{-k(\log_{10}(Re) - \log_{10}(Re)_{onset})}} + \overline{\theta}_{stable} \tag{12}$$

Where $Re_{onset}$ is the value of the function's midpoint, $\overline{\theta}_{unstable} + \overline{\theta}_{stable}$ is the supremum of the values of the function, k is the steepness of the curve, and b is the minimum $\overline{\theta}$ value of the function. Particles with higher area ratio typically have bigger

oscillations with larger $\overline{\theta}$. Low area ratio dendrites and stellar crystals are stable at very high Re, and typically have a smaller $\overline{\theta}$ when unstable.

A motion type parameter $\epsilon$ is calculated using $\theta_{amp}$ and $\theta_{tilt}$ to quantify the spectrum of behaviour from zigzag ($\epsilon = 0$) to spiral ($\epsilon = 1$). $\epsilon$ increases for increasing Reynolds number and dimensionless moment of inertia: spiralling is more likely when both parameters are higher. Particles were observed to exhibit zigzagging behaviour at lower Reynolds number than spiralling,

and some particles (Wang-BBP) were found to exclusively spiral when unsteady. This contrasts with findings from Esteban et al. (2018) and Zhong et al. (2013), who expect spiralling to occur at lower Re and I* than zigzagging.

### 4.2   Implication of results for snowflake modelling and interpretation of observations

It was found that the modal orientation of unstable ice crystals is nonzero (i.e. non-horizontal), Most cloud and radiative transfer models assume ice crystals are oriented horizontally when unstable. Finding a non-zero modal orientation value shows



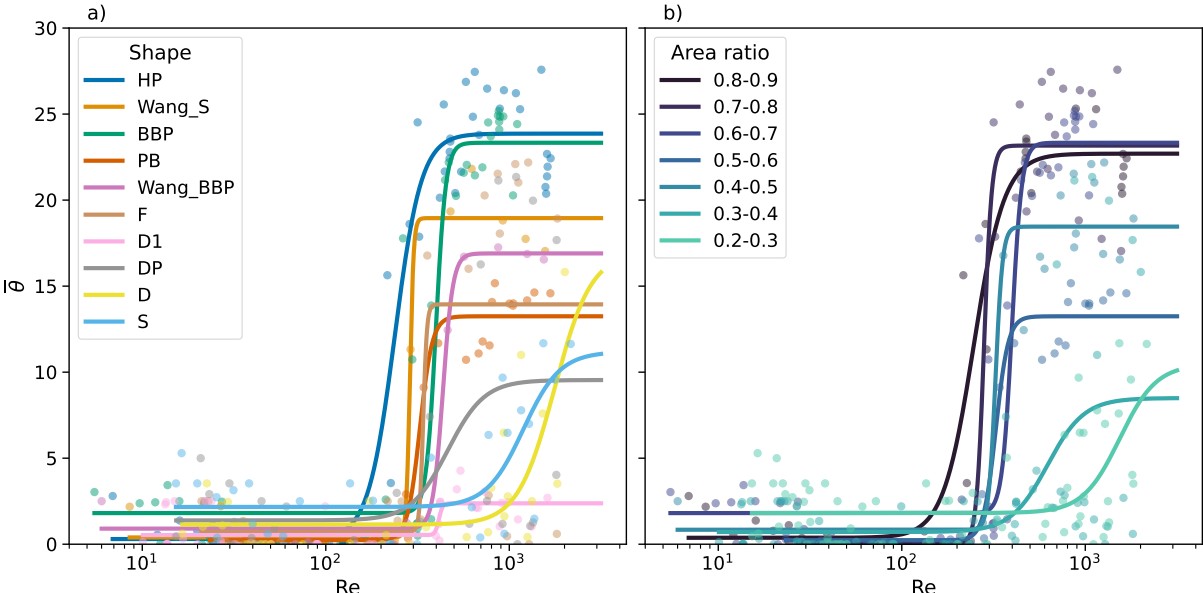

**Figure 17.** Mean inclination angle, $\overline{\theta}$, against Reynolds number, Re, for each experiment (scatter), grouped by particle shape (a) and area ratio (b), with overlaid fitted logistic functions.

this assumption may not always be accurate. Accounting for a non-horizontal modal orientation could improve the realism and accuracy of these models.

Remote sensing techniques that retrieve ice crystal properties through modeling scattering signatures could be improved by incorporating a non-zero modal orientation for unstable crystals. This would lead to better characterization of ice cloud properties from active sensors like radar as well as passive sensors.

In-situ probe measurements of crystal orientation in natural clouds have found preferentially non-horizontal orientations. A non-zero modal value provides a physical explanation for these field observations.

Large fluctuations in velocity for unstable particles imply that a single mean speed cannot be used to approximate the velocity of a single fluttering crystal, and that a spread of velocities must be considered. This should be expected to appear as a broadening of the radar Doppler spectrum at both vertical and horizontal viewing angles. Accounting for this broad spectrum of instantaneous velocities and projected areas of oscillating crystals could improve radar-based retrievals of particle properties.

## 4.3 Limitations of current study, and outlook to future work

Strong turbulence could affect crystal orientation when crystals are large (>1mm), (Garrett et al., 2015). In these cases, strong turbulence is found to widen the distribution of $\theta$ for both stable and unstable particles (Fitch et al., 2021). Our study only considers quiescent conditions, as we want to know under what conditions particle instability still occurs, even without the addition of turbulence.



Although turbulence is typical within convective clouds and at the ground, typical turbulence induced velocity perturbations across the faces of ice crystals within clouds are approximately 50 times smaller than the vertical velocity of the crystal, and other aerodynamic factors are involved (Westbrook et al., 2010). Studies of sun glints have shown that there are many cases in which turbulence does not dominate and conditions can be considered quiescent, such that ice particles have horizontal orientations (Marshak et al., 2017; Varnai et al., 2020). Turbulence does not typically dominate the fall behaviour of very small particles, as the scales of turbulence are not small enough to influence the orientation of the crystals, with only a slight wobble of up to 2 degrees observed in some cases (Sassen, 1980; Klett, 1995).

In literature on circular discs Re is the key control on the onset of unsteadiness, while the parameter I* is argued to modulate the form of the unsteady motion (recall Fig. 2). In our data, it is clear that I* on its own is not the only relevant parameter, or perhaps not even the leading control (see section 3.3.2). Nevertheless we acknowledge that I* in our current experiment is significantly smaller than the case of ice crystals falling in air, largely due to the difference in density between the lab fluid (water) versus the atmosphere (air). To address this, we are currently undertaking a new set of experiments with much lighter analogues falling in air, and will report these results in a future publication.

Many aspects of shape are not covered by area ratio, and other shape parameters could later be explored in addition to area ratio, to capture the full variability of shape parameters. Future work therefore also includes exploration of the impact of shape, with the aim of understanding the influence of experimental conditions on unsteadiness more accurately than the results presented in this study.

Understanding the fall behaviour of ice crystals allows us to further understand the speed at which they grow, fall, and precipitate, allowing this behaviour to be modelled and parameterized more effectively. Further research exploring an even wider range of ice crystal shapes, sizes, and environmental conditions will help build on these findings and advance our overall understanding of ice crystal dynamics within the complex atmospheric system.

*Data availability.*

Data for the case studies and the characteristics of the full data set are available as supplementary material.

*Video supplement.* Videos from the four case studies presented in Section 3.2 are supplementary material.



**Table A1.** Output variables from logistic fits as presented in Figure 17 for each shape

| Shape | $\bar{\theta}_{unstable}$ (°) | $Re_{onset}$ | $k$ | $\bar{\theta}_{stable*}$ (°) |
|---|---|---|---|---|
| HP | 23.6 | 237 | 14 | 0.2 |
| Wang-S | 18.6 | 287 | 100 | 0.4 |
| BBP | 21.5 | 398 | 37 | 1.8 |
| PB | 13.0 | 325 | 30 | 0.3 |
| Wang-BBP | 16.0 | 435 | 37 | 0.9 |
| F | 13.2 | 339 | 100 | 0.7 |
| D1 | 1.85 | 413 | 100 | 0.5 |
| DP | 8.16 | 462 | 10 | 1.4 |
| D | 15.8 | 1763 | 10 | 1.1 |
| S | 9.02 | 1193 | 10 | 2.2 |

**Table A2.** Output variables from logistic fits as presented in Figure 17 by binned area ratio.

| Area Ratio | $\bar{\theta}_{unstable}$ (°) | $Re_{onset}$ | $k$ | $\bar{\theta}_{stable*}$ (°) |
|---|---|---|---|---|
| $0.8 > \& \leq 0.9$ | 22.3 | 247 | 12 | 0.4 |
| $0.7 > \& \leq 0.8$ | 23.2 | 281 | 5 | 0.0 |
| $0.6 > \& \leq 0.7$ | 21.5 | 398 | 37 | 1.8 |
| $0.5 > \& \leq 0.6$ | 13.0 | 326 | 30 | 0.2 |
| $0.4 > \& \leq 0.5$ | 17.6 | 324 | 52 | 0.8 |
| $0.3 > \& \leq 0.4$ | 7.79 | 622 | 10 | 0.7 |
| $0.2 > \& \leq 0.3$ | 8.65 | 1540 | 10 | 1.8 |

**Appendix A: Appendix**

*Author contributions.* JS contributed to conceptualization, data curation, software, formal analysis of the work, and wrote the manuscript draft. CW and TS provided supervision, project administration, and helped write the original manuscript draft. MM produced the dataset and provided data curation. All authors provided conceptualization and review and editing of the manuscript.

*Competing interests.* All authors declare that they have no conflicts of interest.



*Acknowledgements.*  This work was conducted on a PhD studentship through the SCENARIO Doctoral Training Program, funded by NERC, project code F4114950.



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
