# Peer review of "Stable and unstable fall motions of plate-like ice crystal analogues"

_EGUsphere, 2024_

## Author Response (AR1)

**Comments from Reviewer 1:**

*The study by Stout et al. investigates the kinematic behaviour of plate-like ice crystal analogues, as they sediment in a controlled laboratory environment. The study relies on experiments conducted as part of previous published work, and expands upon it by characterizing varying trajectory regimes, and the associated orientation, velocity fluctuations, and projected area fluctuations of the analogues. Among other findings, they show that in certain regimes the commonly adopted assumption of ice crystal orientation being distributed normally around 0° may not be accurate. This finding is highly valuable for a number of sub-disciplines within the cloud physics community, as assumptions on ice crystal orientation are needed for radiative transfer simulations, as well as the interpretation of lidar and radar observables.*

*I find the manuscript to be of high scientific quality: the motivation behind the study is clearly illustrated, the analysis section is preceded by a comprehensive literature review on the topic, the analysis methods are sound, the figures are clear and polished, and the text is overall very clear and well-written. I did not find any major flaws in the manuscript, and I recommend it for publication in ACP after a round of relatively minor revisions. The revisions I request are mainly to include minor missing details, clarify some ambiguities, and improve the clarity of certain portions of the text.*

**General points**

When fitting sine waves to the particles' elevation time series, the goodness of the fit should be discussed. I believe a comprehensive discussion is not necessary, as the assumption is supported by previous studies already cited by the authors. However, I would like the goodness of fit to be quantified and discussed at least for the four case studies. As an example, the author could calculate a RMS deviation between the observed and fitted thetas, possibly normalized by the wave amplitude.

Response:

Thank you for pointing this out – I agree. Thankfully, error analysis was carried out when the fits were first made and were included in the supplementary material as "MSE_a" ("MSE_over_std_a") as the mean-squared-error of the fits to θ (normalised by the standard deviation of θ). Upon re-examination, I realise the supplementary material data is illegible (due to a comma delimitation error).

Actions:
To amend this, I will supply a legible version of the supplementary data, and I have added the following sentence to the manuscript to the second paragraph of section 3.2.2:

"The root mean square error of the rectified sine wave fits to the four case studies is 1.1\degree, 1.3\degree, 2.7\degree, and 0.9\degree for the zigzag, transitional-zigzag, transitional, spiral, cases respectively. The RMSE of all fits to the data are provided in the supplementary data and are 1\degree on average."

And the RMSE values have also been added to the amended supplementary data.

In my opinion the abstract is not very effective at "marketing" the article. The findings of the study are highly relevant for a number of applications, as discussed by the authors in section 4.2, and this could be briefly highlighted in the abstract itself. For example, when mentioning that "the Gaussian model of inclination angle is common in the literature", another sentence could be added clarifying

in what circumstances this assumption is used and in what instances the findings of this study should be taken into account.

Response:

Thank you for saying our findings are highly relevant. This is good advice; we have revised the abstract in line with this suggestion.

Section 4.2: the considerations made here are quite vague, and I believe it would be valuable for the community to include practical recommendations on how to take into account the findings of this study. For example: could the results of this study be directly included in forward simulations to assess the effect of the findings on e.g., radar observations and shortwave reflection? Or is more work needed in this regard? If the results here presented could indeed be directly used in forward simulations, can you describe practical steps on how to do it?

Response:

On reflection, we agree. We have revised this subsection to provide greater clarity on what some of the implications are, and how the results presented here could inform future developments (for example by including them in forward simulations of particle scattering, as the reviewer suggested).

**Minor points**

- Lines 5-6: when mentioning the zig-zag motions it would be helpful to clarify that it occurs in a constant vertical plane, as this aspect was unclear to me when first reading the manuscript.

Response: Added "in a constant vertical plane"

- Lines 21-23: upon first reading the introduction it was unclear to me how the different trajectory regimes could affect the orientation distribution. Since it is key that the reader understands this aspect, I believe it is already worth explicitly explaining it here.

Response: We agree, thank you for pointing this out.

The following sentence has been added: "However, ice crystals exhibit a variety of unstable falling regimes, each corresponding to different distributions of orientations."

- Lines 74-75: upon first reading the article it was unclear to me what "more planar" and "more circular" actually meant. I suggest that the authors rephrase in a clearer way.

Response:

I agree. I have rephrased the sentence to now say:

"Zhong et al. (2011) find that for circular discs, at low Re and I*, the most common behaviour is spiralling, whereas at high Re and I* zigzagging behaviour is most common, with transitional behaviour occurring at intermediate I* and Re." which I hope is clearer. The areas of expected regime are labelled on the figure.

- Section 1.2.2: it would be helpful for the reader to have the moment of inertia of a thin disc and of a sphere included here for reference. Alternatively the authors could refer to the precise equations in a classical mechanics textbook.

Response:

A reference to a classical mechanics textbook has been provided at the end of section 1.2.2.

- Line 104: I suggest that here it is specified that the range is indicated by the hatched area in the figure.

Response: Good suggestion. We have added the clause "The hatched region of" to the beginning of the sentence.

- Line 120: it is unclear to me what "orientation model for falling particle" means, and in what ways it is different from the "models of particle orientation" mentioned in the previous sentence. Can the authors briefly provide more details on the study by Klett (1995) and the assumptions made therein?

Response:

Apologies for the confusion and thank you for pointing this out. The same meaning was intended for each phrasing: the distribution function of inclination angle.

We have changed both instances to now say "models of particle orientation distribution"

- Lines 123-124: I find the sentence "however, these indirect measurements lack the spatial resolution necessary to represent microphysical processes without making many assumptions" very unclear and vague. Can the authors concisely explain the assumptions used by Melnikov and Straka and why these are unrealistic?

Response: Apologies for the lack of clarity here. We had not intended to imply any remote sensing studies were unrealistic.

This sentence has been changed to now say: "However, remote sensing is an indirect measurement rather than a direct observation of the fall motion."

- Lines 147-148: since the pendulum assumption is key to the analysis presented later in the article, I believe it would be helpful for the reader here to explicitly describe the functional form of an orientation distribution associated with such pendulum motion.

Response:

The pendulum analogy in this paper serves purely as a conceptual device to help readers visualise the modes of motion and is not necessarily key to any of the analysis or results. We do not believe it it unnecessary to explore this in depth here, as our focus remains on the real measurements rather than theoretical analogues.

To address this point, we have added this point to the paper when discussing figure 4: "A conical pendulum characteristically traces out a circle in the horizontal plane, akin to spiralling cases, which also trace out a circle in the horizontal plane. Similarly, a planar pendulum serves as an analogy for the zigzagging motion, as they are both constrained to movement in a single plane, tracing out a line in the horizontal plane."

For the reviewer's reference:
The inclination angle distribution of a planar pendulum swinging between 0 and 5 degrees is represented by y = |5sinx|. A perfect conical pendulum exhibits a single inclination angle, while a spherical pendulum may feature diverse sinusoidal distributions. See below for examples of planar (left) and spherical pendulum (right).

[Figure]

- Line 241: how is unstable periodic detected? Are all non-stable motions classified as unstable periodic? This should be clarified in the text.

Response:

Everything classified in this paper as unstable is also classified as periodic. There was no complex tumbling, or behaviour with any complex, non-sinusoidal motion in it.

There are some experiments in the study which have aperiodic motions, or motions that have other, additional modes of motion on top of the sine wave behaviour. Spiralling motion is periodic because it periodically rotates, zigzagging motion is periodic because it periodically swings. For example, the transitional case has another mode of motion (the sine wave grows in amplitude). All cases are periodic in some sense, but the number of frequencies is sometimes above 1.

The mean RMSE for all sine wave fits is about 1 degree.

- Line 250 / Fig. 3: I assume that the categorization into zig-zag / transitional / spiral here is made using the epsilon parameter introduced later in the text? Or was it by visual inspection of the cases? This should be briefly stated in the text.

Response:

Thank you for pointing out our lack of clarity here. They were picked by visual inspection of the cases, and epsilon was calculated afterwards. I will forward reference the calculation of epsilon straightaway, though.

The introductory sentences have been amended to now read: "These cases were picked by visual inspection as characteristic types of behaviour. In this section, we will quantitatively describe the 4 case studies, and then objectively classify their motion based on inclination angle in Section \ref{Section:epsilon}."

Section 3.2.5 has also been amended to clarify the relevance of epsilon.

- Lines 391-392: This sentence reads like something is missing, what is separated? Into what? I assume the authors mean something along the lines of: "… separates the cases into the different particle shapes and the zigzag and spiraling regimes, to demonstrate their impact on the distributions."

Response: Yes – this must have been accidentally deleted.

The sentence now reads: "Figure 9 displays the distributions of inclination angle across Re, excluding stable experiments, and separates the cases by particle shape and $\epsilon$ greater than or less than 0.5 to demonstrate the impact of $\epsilon$ on the distributions."

- Lines 502-511: I have the impression that these lines don't belong in this section. The current section focuses on summarizing the findings, while these lines seem to introduce a new portion of the analysis. I suggest that these lines are moved earlier in the text, possibly in a new section. After applying this change, the current section could be simply named "summary", as the current title is vague and not very indicative for the reader.

Response:

Thank you for pointing this out – you are correct. This section has been moved into the results section, and the subsections of the discussion have been retitled with simpler titles (e.g. "Summary").

- Line 532: What does "strong turbulence" mean quantitatively? Can you quantify in terms of eddy dissipation rate and mention cloud types where turbulence could be considered strong?

Response:

No direct measurements of turbulence were obtained during the field experiments described in Garrett (2015) or Fitch (2021), they define "strong" as being in the upper quartile of their estimates of turbulence (using gusts and average wind speed as a proxy). A mention of the cloud types (convective and close to the ground) is described in the paragraph below.

Action: The eddy dissipation rate from Klett 1995 has been added in an appropriate place in the section.

**Stylistic/technical corrections**

- I believe that throughout the whole text some figure references are wrong. For example, throughout section 3.2.2 I believe Fig. 5 is referenced instead of Fig. 6, and at line 361 Fig. 7 is referenced instead of Fig. 8. I suggest that the authors double check all figure references.

Response: Thank you for pointing this out. Figure 5 was incorrect (a duplicate of Figure 6) – this was amended during the comment period.

- I'm wondering if the acronyms used for the different particle analogue shapes could be replaced with more intuitive phrases (e.g., HexPl instead HP, FerDen instead of F, DenAPl instead of DP). As a reader it is quite hard to remember all of them, and I often found myself having to go back to Table 1. In my opinion this change would improve the ease of interpretation of the figures, but of course it is not a necessary change.

Response: The notation was intended to match the notation used in the supplementary data from the TRAIL papers (McCorquodale 2021). You're right that new notation could be more intuitive, but keeping consistent with previous work felt more sensible here.

- St_theta is sometimes denoted as St, the notation should be kept consistent throughout the whole text and all the figures.

Response: Thank you for pointing this out. One instance in the text was corrected from St to St_theta. There is a paragraph which references "St" without any subscript, as this is referring to the use of Strouhal number in another paper, and not our particular usage of St (which is always followed by a subscript).

Figure 13 has also been amended to include the subscript.

- Line 18: the phrase "produce a trajectory" sounds odd to me, maybe it could be replaced with "follow a trajectory"?

Response: We agree – We have changed "can produce fluttering, spiralling, and tumbling trajectories" to "can exhibit fluttering, spiralling, and tumbling motions."

- Line 34: Doppler should be capitalized

Response: Corrected, thank you.

- Line 39: The reference to Platt (1977) should be in parentheses

Response: Corrected, thank you.

- Line 94: I assume the authors meant to refer to I_a?

Response: Yes – this sentence was slightly clunky and repeated itself in the next sentence (which properly describes I_a), so this clause was removed.

- Line 464: "Fig." is missing

Response: Corrected, thank you.

**Comments from Reviewer 2:**

The authors reported observed falling motions of plate-like ice crystal analogues. It adds data to fill up the parameter space. The non-zero mode of the inclination angle is not necessary a new idea, but previous works are based on MASC in some air motion (not only Garrett et al. 2015 but also Jiang et al. 2019), so there is value in providing a benchmark behavior in quiescent fluid. I also found the results regarding the broad distribution of instantaneous vertical velocity interesting.

I would recommend this manuscript for publication on ACP after the following comments are addressed.

Regarding writing, there is huge room to reduce redundancy and improve clarity.

**Main comments:**

The framework (theta and phi) is insufficient to describe the particle orientation. For simplicity, refer to the first plot in https://en.wikipedia.org/wiki/Euler_angles. If I understand correctly, theta and phi are equivalent to beta and alpha, respectively, in Wikipedia's illustration. Then another angle

(gamma in this illustration) is required to describe the orientation. The classification is confusing due to the missing angle. For the motion of the spiralling case (see animation in the supplementary materials), if one follows the tip of one branch, it is rotating quite slowly. This is not the case for the two transitional cases, especially the second. In the text, the rotation around the vertical axis is commented on for the transitional cases but omitted for the spiralling cases. It could have been systematically described with Euler angles. The analogue to plane and conical pendulums suffers from the similar issue, that is, it also misses degree of freedom in the true orientation evolution. Please justify the omitting of degree of freedom or add the results regarding this additional angle.

We believe the reviewer may have slightly misinterpreted the text – this paper does not intend to discuss Euler angles themselves, and instead derives inclination angle and azimuthal angle from the Euler angles. These derived angles are more relevant for atmospheric applications.

That is, the Euler angles are useful for reconstructing particle trajectory, but the results of our study are of most relevance to applications for which a detailed trajectory reconstruction is not relevant. In these applications, we believe data regarding the inclination and azimuthal angles will be most useful for the academic community.

To reduce the potential for confusion amongst readers, clarifications regarding this point have been added to section 2, as follows:

*"The Trajectory Reconstruction Algorithm implemented through Image anaLysis (TRAIL) then produced digital reconstructions of the trajectory and orientation of the particle in free fall. The orientation of the particles was reconstructed using a set of Euler angles. [...]*

*This data, referred to as TRAIL, provides time series of the 3D positions and orientation of the falling analogues, from which the 3D velocity vectors at each time step can be derived. The reconstructed orientations, described by the Euler angles, further permit the calculation of the inclination angle, $\theta$, which is more widely used in atmospheric applications. [...]"*

For clarity of the reviewers, the original Euler angles provided by the TRAIL algorithm are shown below:

[Figure]

These Euler angles are not the same as the derived angles presented in the paper, although for clarity we note that the intrinsic rotation mentioned by the reviewer above is indeed the same as gamma here.

**Minor Comments**

L101: Would you please be explicit about the mass-diameter relationship (from Nakada and Terada 1935) and I* estimation formula (from Kajikawa 1992)? Why do you use these two formulae to provide reference

The I* estimation formula from Kajikawa 1992 was chosen as it offers a practical advantage over more complicated relationships, as it only requires the mass and diameter of the crystal (no detailed knowledge of the full distribution of mass around the snowflake is needed).

While alternative mass-dimension relationships could have been chosen, the mass-diameter relationship from Nakada and Terada (1935) was chosen arbitrarily as an illustration of the order of magnitude approximation for reference, so that readers have an estimate of the possible range of values.

To help with clarity: the full calculation of I* example range has been added to the appendix.

L237: Do you mean all three Euler angles fluctuate less than 2.5 deg? You only defined two of them in Figure 1. Is the third one measured?

Yes - all three Euler angles are reconstructed. In order for a particle to be classified as stable, the Euler angles that describe rotation about the a- and a'-axes (see figure 1) must fluctuate by less than $\pm2.5\degree$ across the measurement region. These Euler angles correspond to a particle that falls in a constant orientation, which corresponds to a near-zero inclination angle.

We have revised this description within the manuscript, and combined with the changes made to section 2, we believe the revised description clarifies this point.

L252: Also, low aspect ratio D

Thank you for pointing this out, particle D has been added to the list in this line.

L252-255: The logic in these two sentences is confusing. Is increased CD a cause or a result of the unstable motion?

The relationship between the change in drag coefficient and unstable motion is complex; as the Reynolds number increases wake instability arises, which is coupled with the onset of unsteady motions. The change in wake structure is also thought to result in an increase in drag coefficient. In summary, these phenomena are coupled; a full explanation of these phenomena is beyond the scope of this study. Therefore, we have provided references within the narrative to relevant studies that interested readers can refer to, as follows:

"Previous studies report an increase in drag coefficient when planar particles fall unsteadily (McCorquodale and Westbrook, 2021b). That is, the onset of unsteady motion is coupled with a change in wake structure (Zhong et al., 2011; Tagliavini et al., 2021b, Nettesheim and Wang, 2018), which in turn influences the drag coefficient. This change in $C_D$ is more pronounced when area ratio is high than when it is low (McCorquodale and Westbrook, 2021b), suggesting that unsteadiness is less vigorous in particles with low area ratios, such as dendrites."

Figure 5: In the corrected version of this figure, would you please show the fitted theta together with the raw theta (which I think is what is currently shown)?

Yes, only real values of theta were shown. We have updated Figure 5 and its caption to now show the figure below:

[Figure]

L311: Which figure you are referring to for constant dphi/dt? At least it is not clear in Figure 6. Please clarify.

Apologies - dphi/dt is not constant in the other parts of the Figure, but is for panel d. I have added in a reference: "(as seen in Fig. 6d)" – the constant dphi/dt should be clear in Figure 6d. More references to the other subplots in Figure 6 have also been added.

L316: "the arms of the crystal": Is this the same as the a-axis? Or is it the same as "branch"?

All instances of "arms" are now changed to "branches." The a-axis of the crystal is aligned with a branch, but not all instances of branches are the same as the a-axis.

L338: "the angle of the particle": What is this?

Thank you for pointing out the lack of clarity here. I have added the word "inclination"

Figure 8: It makes sense that, as suggested in the text, the thickness of a particle causes the disagreement between cosine theta and the ratio between the instantaneous and max projected areas. Is there any analysis of actual images that supports this argument? Also, the caption says "modelled projected areas as seen from below". Please explain how it is modelled.

Thank you for bringing this point to our attention – there was previously ambiguity relating to the use of the word "modelled"; we were remiss not to use the "reconstructed". That is, the projected area is determined from the digital reconstruction of the particle by projecting the vertices of the reconstructed particle onto a horizontal plane. We have amended this term within the caption of figure 8.

We note that it would be unreliable to perform analysis based on the raw images, since the particle is not strictly seen from directly beneath by the camera (since the camera may not be perfectly vertical and the particle may not be perfectly centred over the camera); we mitigate these issues through the use of the digital reconstruction.

The analysis of the reconstructed trajectories does support this argument, since (at times) the observed projected area exceeds that of the projected area of the particle when horizontal (i.e. the ratio in figure 8(b) exceeds 1); this can only be achieved through the influence of the thickness of the particle.

L358: "as drag is minimised in the horizontal plane": What do you mean by this?

Changed to "drag is minimised in the vertical direction" – thank you for pointing out the lack of clarity here.

L376: What is "v" in the denominator?

Apologies – here we meant mean vertical fall velocity. All instances of u or v in the equations and text that refer to mean vertical velocity now are corrected to "V_mean"

Figure 9: What are the values before and after the +/- sign for mean theta amp and mean theta tilt? Also, it helps readers if you can label the distributions with dominant falling sub-types and numbers of cases.

Thank you for pointing out the lack of clarity - Those values correspond to the mean observed values of theta_amp and theta_tilt and the standard deviations of those mean observed values.

The caption of Figure 9 has been changed to include the line "Reported values are the mean ($ \pm$ standard deviation of the mean) for each set of distributions."

The most prevalent regime of motion at that Reynolds number should be evident from this figure as the motion types are already split by epsilon here. The motion types are labelled explicitly by individual case in Figure 3: adding even more labels to Figure 9 may make it overly cluttered.

Figure 13: The high epsilon cases in Panel c clearly are not well described by a linear fit. Have you tried to link V/Vmean for these cases to other potential controlling variables?

Thank you for pointing this out – the fit was created and added to a different iteration of the data, it has now been removed.

Yes – All the variables mentioned in this study were explored, but no other results were worth showing (as results were nonlinear or unconvincing).

Subsection 3.3.4 Strouhal numbers: Is there any relationship between the two St numbers and area ratio?

Not that we have managed to ascertain. Here are the plots of Strouhal numbers versus area ratio:

[Figure]

L449: Why do you use data for cylinder as reference while all particles tested are plate-like? Hashino et al. 2014 (doi:10.1016/j.atmosres.2014.07.003) has some simulations on vortex structure around falling plates and may be a useful reference for some of your parameter space.

We chose to use cylinders as a reference in this section due to a lack of available literature discussing the frequency of eddy shedding behind hexagonal plates. We appreciate the provided reference (Hashino et al., 2014) and acknowledge its potential relevance for future investigations. However, since it does not directly address the specific parameters or phenomena explored in our study, we have opted not to cite it in this manuscript.

While Hashino et al. (2014) examines flow characteristics and torque distribution around ice columns and hexagonal plates, it does not delve into the frequency of eddy shedding or the parameters central to our analysis, such as I* or onset of different motion subtypes at varying Reynolds numbers. Therefore, we believe our decision aligns with the focus and scope of our research.

L462: Doesn't it take both theta_amp and theta_tilt to decide whether a particle is falling in the transitional sub-type?

Yes, you are correct. We have added in the clause: "For particles that are already spiralling,"

L479: In the animation for the spiralling case, it seems that the center of mass is rotating around the y-axis but the particle is not rotating around its vertical axis. See main comments. Please clarify.

A point of clarification has been added to section 4.1, as follows:

"*Spiralling particles rotate steadily around the vertical axis at constant $d\phi/dt$. That is, the rotation of spiralling planar particles does not result from a rotation around the c-axis of the particle (see figure 1); rather, periodic rotations of equal amplitude, but phase difference of 90$^o$, occur about the a-axis and a'-axis such that $\theta$ is approximately constant. This rotation of the particle about the a- and a'-axes causes the particle's centre of mass to trace a circular path*."

We believe this clarification, coupled with the clarifications added to sections 2 and 3, removes any potential ambiguity.

L495: Please specify that this is consistent with previous work instead of a new result.

Added: "consistent with previous work \citep{Tuleja2007}"

L496: What about spiralling or other falling sub-types?

"The description of Strouhal numbers for theta for other types of motion (i.e., the wobble rate of spiralling particles) was omitted from the paper's summary due to their limited relevance to broader scientific discourse. Further elaboration can be found in the dedicated section on Strouhal numbers.

TECHNICAL ISSUES:

Figure 1: the language (an observer "facing parallel" to some plane) is confusing; Panel b does not look like that the observer is facing the a-y plane (actually Panel a looks like it)

Thank you for pointing out this lack of clarity, this has been amended to: "The views provided are for an observer facing (a) the c-y plane (b) the a'-y plane (c) the x-z plane."

Thank you for pointing out the error in the description of panel b, which is indeed the a'-y plane and not the a-y plane.

Table 1: This table defines most acronyms/abbreviations (TRAIL, CD, PB, and so on) and should be moved to before the first time they are used (mostly in Figure 2).

The following has been added to the caption of Figure 2:

"Acronyms in legend refer to the shapes Table \ref{tab:shapes}." Understanding the acronyms is not key to understanding the figure, as the shapes are still described in the caption.

Section 1.2: It is confusing when the authors simultaneously used terms like "circular discs", "planar discs", and "thin discs", and mixed references for idealized discs and real ice crystals all in one subsection for "circular discs".

Apologies for the lack of clarity here. The instance referring to "planar discs" has been changed to "thin circular discs" and all instances of "thin discs" have been changed to either "thin circular discs" or simply "circular discs" as disc already implies that the shape is thin.

L80: Please be specific about which subsection of Pruppacher and Klett the authors are referring to.

We have added "see chapter 10" to the in-line citation.

L94: There is no 'I' in Eq. 3. Please clarify.

Thank you for pointing this out. This was clarified with Reviewer 1 above. The sentence now reads: "where D is the maximum dimension of the particle. The moment of inertia for rotation around the 3 principal axes of the crystals is calculated, where $I_{a}$ is the smallest of these three moments, aligned in the a-axis of the crystal (Fig. \ref{fig:crystalaxes}) \citep{Kajikawa1992}."

L236: Why is there so much description of unsteady motion in this subsection titled "Crystals which fall steadily"

This section discusses when steadiness occurs, and therefore must discuss when steadiness stops occurring (i.e., the onset of unstable motions).

L243: Unit for moment of inertia seems missing, or do you mean the dimensionless one?

Yes – thank you for pointing this out – I have added the word "dimensionless".

L245 and several other places: "onset of stability": do you mean "onset of unstable motion"?

Yes, thank you for pointing this out – this has been amended to your suggestion.

L265: Please refer to plots (panels) when describing each falling sub-type.

References to figure 6 subplots have been added accordingly.

L311: "dphi/dt is constant such that it precesses at a constant rate": this is redundant.

Thank you - "such that it precesses at a constant rate" has been removed.

Words like "rock" and "see-saw" are too informal.

Agree – I have amended this by replacing see-saw and rock with pivot or swing.

Figure 13: Panel c: missing operator before 0.019?

The line of best fit and its equation was removed so this no longer needs amending.

---

## Author Response (AR2)

Dear Ann Fridlind,

Thank you for your feedback and the opportunity to revise our manuscript. We appreciate the insightful comments and suggestions provided by you and the reviewers, which have greatly helped us improve the quality of our work.

The requested text revisions have been made as follows:

Regarding unstable motion:

- The following text has been added to Section 3.2.2, after the introduction of the sine wave fit (line 294):
  "All unstable motion presented in this study was observed to be periodic and is approximated through equation 4. We observed no complex tumbling, chaotic fluttering, or behaviour with significantly non-sinusoidal motion to it. However, we note that there are some experiments in the study in which additional (weaker) modes of oscillation, in addition to the primary frequency, seem to be present. For example, the transitional case in Fig. 5c has an amplitude that fluctuates slightly in time, at a lower frequency than the primary mode of oscillation captured by the simple single-frequency fit. We did not attempt to capture these finer details in our fitting procedure."

Regarding particle reconstruction the following has been added to the section on projected areas, line 384:

- "In fact, the observed projected area occasionally exceeds that of the projected area of the particle when horizontal (i.e., the ratio in Figure 8b slightly exceeds 1): this can only be achieved through the influence of the finite thickness of the particle."

Re I*: The following has been added as a note at the end of the Supplementary Material:

- "Although alternative mass-dimension relationships exist, the mass-diameter relationship from Nakaya and Terada (1935) was chosen arbitrarily to illustrate the order of magnitude approximation for reference, providing an estimate of the possible range of values. The I* estimation formula from Kajikawa (1992) was selected for its practical advantage over more complex relationships, requiring only the mass and diameter of the crystal without detailed knowledge of the full mass distribution around the snowflake."

Re Figure 13: The following sentence has been added to the end of Section 3.3.3 (Velocity fluctuations) (line 470):

- " Other relationships between the variables mentioned in this study were also explored; however no clear patterns or simple relationships were evident."

Re Strouhal numbers: The following sentence has been added to the second paragraph of Section 3.3.4 (Strouhal numbers) (line 485):

- "No systematic relationship was found between either Strouhal number ($St\_theta$ or $St\_phi$) and area ratio."

Re use of cylinders as reference:

- Upon further consideration, we have omitted the paragraph on vortex shedding and its relation to Strouhal numbers. The connection to vortex shedding behind cylinders was deemed too speculative. To maintain clarity, we have decided to report the results directly and avoid potential confusion in the text.

In addition, the accidental inclusion of coloured text in the manuscript was removed and replaced with black text.

Please let us know if there are any further changes or clarifications needed. Thank you for your time and consideration.